# Evolutionary transcriptomics implicates new genes and pathways in human pregnancy and adverse pregnancy outcomes

Katelyn Mika[1,2†], Mirna Marinić[1,2†], Manvendra Singh[3†], Joanne Muter[4,5], Jan Joris Brosens[4,5], Vincent J Lynch[6*]

[1]Department of Human Genetics, University of Chicago, Chicago, United States; [2]Department of Organismal Biology and Anatomy, University of Chicago, Chicago, United States; [3]Department of Molecular Biology and Genetics, Cornell University, Chicago, United States; [4]Tommy's National Centre for Miscarriage Research, University Hospitals Coventry & Warwickshire, Coventry, United Kingdom; [5]Division of Biomedical Sciences, Clinical Sciences Research Laboratories, Warwic Medical School, University of Warwick, Buffalo, United States; [6]Department of Biological Sciences, University at Buffalo, Buffalo, United States

*For correspondence: vjlynch@buffalo.edu

†These authors contributed equally to this work

Competing interest: The authors declare that no competing interests exist.

**Abstract** Evolutionary changes in the anatomy and physiology of the female reproductive system underlie the origins and diversification of pregnancy in Eutherian ('placental') mammals. This developmental and evolutionary history constrains normal physiological functions and biases the ways in which dysfunction contributes to reproductive trait diseases and adverse pregnancy outcomes. Here, we show that gene expression changes in the human endometrium during pregnancy are associated with the evolution of human-specific traits and pathologies of pregnancy. We found that hundreds of genes gained or lost endometrial expression in the human lineage. Among these are genes that may contribute to human-specific maternal–fetal communication (*HTR2B*) and maternal–fetal immunotolerance (*PDCD1LG2*) systems, as well as vascular remodeling and deep placental invasion (*CORIN*). These data suggest that explicit evolutionary studies of anatomical systems complement traditional methods for characterizing the genetic architecture of disease. We also anticipate our results will advance the emerging synthesis of evolution and medicine ('evolutionary medicine') and be a starting point for more sophisticated studies of the maternal–fetal interface. Furthermore, the gene expression changes we identified may contribute to the development of diagnostics and interventions for adverse pregnancy outcomes.

## Introduction

Evolutionary changes in the ontogeny of anatomical systems are ultimately responsible for their functional conservation and transformation into new tissue and organ systems (novelties) with new physiological functions that are outside of the range of the ancestral ones (innovations). These same evolutionary and developmental histories limit (constrain) the range of genetic and environmental perturbations those physiological functions can accommodate before leading to dysfunction and disease (i.e., their reaction norms). Evolution of the structures and functions of female reproductive system and extraembryonic fetal membranes, for example, underlie the evolution of pregnancy (*Armstrong et al., 2017*; *Hou et al., 2009*; *Kin et al., 2015*; *Lynch et al., 2015*; *Lynch et al., 2008*) and likely adverse pregnancy outcomes such as infertility (*Cummins, 1999*), recurrent spontaneous

**eLife digest** Pregnancy is a complicated process. It has three phases: the body recognizes the embryo, it maintains the pregnancy, and finally, it induces labor. These stages happen in all mammals, but the details are different in humans. Human pregnancy and labor last longer. We menstruate. Our placentas invade deeper into the uterus, and the cues that signal pregnancy is done and induce labor are different than in most other mammals. We are also more likely to have pregnancy complications, including infertility, a dangerous rise in blood pressure called preeclampsia, and premature birth. The reasons for these differences are unknown.

Human pregnancy relies on close communication between the placenta and the uterus. The immune system must allow the placenta to grow large enough to support the developing embryo, and blood vessels need to adapt to supply gases and nutrients and to remove waste. Understanding how the genes used by the human uterus are different to those used in other species could help explain why human pregnancies are so unusual.

Mika, Marinić et al. compared the genes used by the pregnant human uterus to those used in 32 other species, including monkeys, marsupials and other mammals, birds, and reptiles. The analysis revealed that the humans use almost a thousand genes that other animals do not. These genes have roles in the invasion of the placenta, the growth of blood vessels, and control of the immune system. Several have links to the hormone serotonin, which had not been connected with the uterus before. Mika, Marinić et al. suggest that it might control the length of pregnancy, the timing of labor, and communication between parent and baby.

The genes identified here provide a starting point for further investigation of human pregnancy. In the future, this may help to prevent or treat infertility, preeclampsia, or premature birth. A possible next step is to examine our closest living relatives, the great apes. Performing similar experiments using tissues or cells from chimpanzees, gorillas, and orangutans could reveal more about the genes unique to human pregnancy.

---

abortion (*Kosova et al., 2015*), preeclampsia (*Carter, 2011*; *Crosley et al., 2013*; *Elliot, 2017*; *Rosenberg and Trevathan, 2007*), and preterm birth (*LaBella et al., 2020*; *Marinić et al., 2021*; *Plunkett et al., 2011*; *Swaggart et al., 2015*). Thus, reconstructing the evolutionary and developmental history of the cells, tissues, and organs involved in pregnancy may elucidate the ontogenetic origins and molecular etiologies of adverse pregnancy outcomes.

Extant mammals span major stages in the evolution and diversification of pregnancy, including the origins of maternal provisioning (matrotrophy), placentation, and viviparity (*Behringer et al., 2006*; *Freyer et al., 2003*; *Freyer and Renfree, 2009*; *Hughes and Hall, 1998*; *Renfree, 1995*; *Renfree and Shaw, 2013*). Eutherian mammals have also evolved a complex suite of traits that support prolonged pregnancies such as an interrupted estrous cycle, maternal recognition of pregnancy, maternal–fetal communication, immunotolerance of the antigenically distinct fetus, and implantation of the blastocyst into maternal tissue (*Abbot and Rokas, 2017*). There is also considerable variation in pregnancy traits within Eutherians. Catarrhine primates, for example, have evolved spontaneous decidualization (differentiation) of endometrial stromal fibroblasts (ESFs) into decidual stromal cells (DSCs) under the combined action of progesterone, cyclic adenosine monophosphate (cAMP), and other unknown maternal signals (*Carter and Mess, 2017*; *Gellersen et al., 2007*; *Gellersen and Brosens, 2003*; *Kin et al., 2016*; *Kin et al., 2015*; *Mess and Carter, 2006*), deeply invasive interstitial hemochorial placentas (*Carter et al., 2015*; *Pijnenborg et al., 2011a*; *Pijnenborg et al., 2011b*; *Soares et al., 2018*), menstruation (*Burley, 1979*; *Emera et al., 2012b*; *Finn, 1998*; *Strassmann, 1996*), and a unique but unknown parturition signal (*Csapo, 1956*; *Csapo and Pinto-Dantas, 1965*). Humans have evolved interstitial trophoblast invasion, in which the blastocyst is embedded and encased entirely within the uterine endometrium (*McGowen et al., 2014*; *Norwitz et al., 2001*; *Salamonsen, 1999*), shorter interbirth intervals (*Galdikas and Wood, 1990*), and longer pregnancy and labor (*Bourne, 1970*; *Keeling and Roberts, 1972*) than other primates. Humans also are particularly susceptible to pregnancy complications such as preeclampsia (*Crosley et al., 2013*; *Elliot, 2017*; *Marshall et al., 2018*), and preterm birth (*Phillips et al., 2015*; *Rokas et al., 2020*; *Wildman et al., 2011*) than other primates.

Gene expression changes ultimately underlie the evolution of anatomical structures, suggesting that gene expression changes at the maternal–fetal interface underlie these primate- and human-specific pregnancy traits. Therefore, we used comparative transcriptomics to reconstruct the evolutionary history of gene expression in the pregnant endometrium and identify genes that gained ('recruited genes') or lost endometrial expression in the primate and human lineages. We found genes that evolved to be expressed at the maternal–fetal interface in the human lineage were enriched for immune functions and diseases such as preterm birth and preeclampsia, as well as other pathways not previously implicated in pregnancy. We explored the function of three recruited genes in greater detail, which implicates them in a novel signaling system at the maternal–fetal interface (*HTR2B*), maternal–fetal immunotolerance (*PDCD1LG2*), and remodeling of uterine spiral arteries and deep placental invasion (*CORIN*). These data indicate that explicit evolutionary studies can identify genes and pathways essential for the normal healthy functions of cells, tissues, and organs, and that likely underlie the (dys)function of those tissue and organ systems.

## Results

### Endometrial gene expression profiling and ancestral transcriptome reconstruction

To identify gene expression gains and losses in the endometrium that are phylogenetically associated with derived pregnancy traits in humans and catarrhine primates, we assembled a collection of

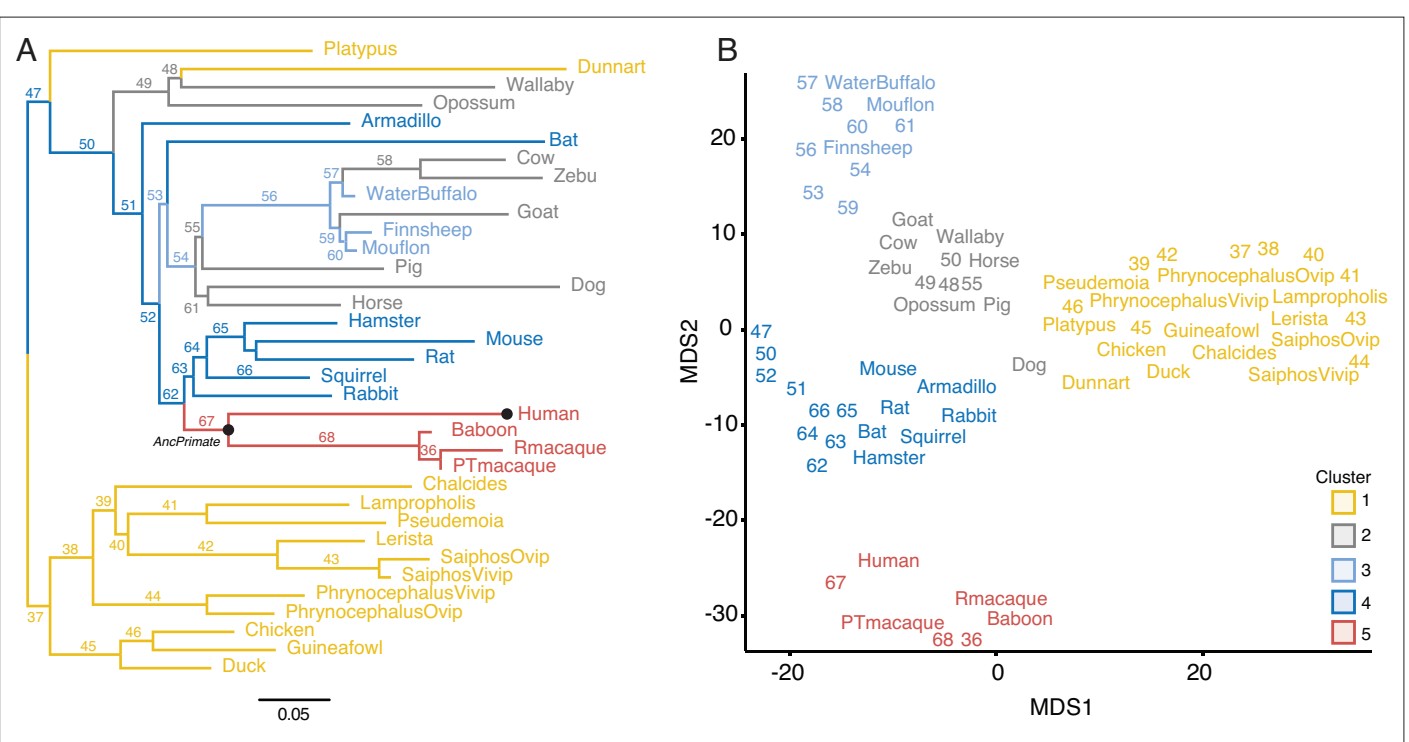

**Figure 1.** Gene expression profiling at the amniote maternal–fetal interface. (**A**) Amniote phylogeny with branch lengths drawn proportional to the rate of gene expression gains and losses (per gene). The ancestral primate (AncPrimate) and human nodes are indicated with black circles. Internal branches are numbered. (**B**) Multidimensional scaling (MDS) plot of binary encoded endometrial gene expression data for extant and ancestral reconstructed transcriptomes. Transcriptomes are colored by their group membership inferred from *K*-means clustering with *K* = 5. Numbers indicate internal branches from panel A.

The online version of this article includes the following figure supplement(s) for figure 1:

**Source data 1.** Species names (common and binomial), genome annotations used for RNA-Seq analysis, and parity mode.

**Source data 2.** Gene expression matrix and ancestral reconstruction results.

**Source data 3.** Binary encoded matrix of gene expression from extant and ancestral reconstructions used to generate *Figure 1B*.

**Figure supplement 1.** Binary encoding of gene expression data reduces noise.

transcriptomes from the pregnant or gravid endometrium of 20 Eutherian mammals, including human (*Homo sapiens*), baboon (*Papio anubis*), Rhesus monkey (*Macaca mulatta*), and Pig-Tailed macaque (*Macaca nemestrina*), three Marsupials, platypus, three birds, and six lizard species, including species that are both oviparous and viviparous (*Figure 1* and *Figure 1—source data 1*). The complete dataset includes expression information for 21,750 genes and 33 species, which were collected at different gestational times from early- to midpregnancy, by multiple labs, and sequencing methods. Thus, differences in transcript abundance between samples may reflect biological differences in mRNA abundances between gestational ages or species, differences in sequencing protocols, or other technical factors unrelated to the biology of pregnancy (i.e., batch effects). Therefore, we transformed quantitative gene expression values coded as transcripts per million (TPM) into discrete character states such that genes with TPM ≥2.0 were considered expressed (state = 1), genes with TPM <2.0 were considered not expressed (state = 0), and missing genes coded as unknown (?; *Box 1*). Consistent with significant noise reduction, multidimensional scaling (MDS) of species based on gene expression levels (TPMs) was essentially random (*Figure 1—figure supplement 1A*), whereas MDS of the binary encoded dataset grouped species by phylogenetic relatedness (*Figure 1—figure supplement 1B*).

Next, we used the binary encoded dataset to reconstruct ancestral transcriptomes and trace the evolution of gene expression gains (0 → 1) and losses (1 → 0). Ancestral states were inferred with the empirical Bayesian method implemented in IQ-TREE 2 (*Minh et al., 2020*; *Nguyen et al., 2015*) using the species phylogeny (*Figure 1A*) and the GTR2 + FO + R4 model of character (*Soubrier et al., 2012*). Interested readers are referred to other publications for more information about ancestral reconstruction methods (*Joy et al., 2016*; *Pauling et al., 1963*). Internal branch lengths of the gene expression tree were generally very short while terminal branches were much longer, indicating pronounced species-specific divergence in endometrial gene expression (*Figure 1A*). MDS of extant and ancestral transcriptomes (*Figure 1B*) generally grouped species by phylogenetic relationships, parity mode, and degree of placental invasiveness. For example, grouping platypus, birds, and reptiles (cluster 1), viviparous mammals with noninvasive placentas such as opossum, wallaby, and horse, pig, and cow (clusters 2 and 3), and Eutherians with placentas such as mouse, rabbit, and armadillo (cluster 4). Human, baboon, Rhesus monkey, and Pig-Tailed macaque formed a distinct group from other Eutherians (cluster 5), indicating that catarrhine primates have an endometrial gene expression profile during pregnancy that is distinct even from other Eutherians (*Figure 1B*).

## Gain and loss of signaling and immune regulatory genes in humans

We identified 923 genes that gained endometrial expression in the human lineage with Bayesian posterior probabilities (BPPs) ≥0.80 (*Figure 2—source data 2*; *Box 2*). These genes are enriched in 54 pathways, 102 biological process Gene Ontology (GO) terms, and 91 disease ontologies at a false discovery rate (FDR) ≤0.10 (*Figure 2*). Among enriched pathways were 'GPCRs, Class A Rhodopsin-like', 'Signaling by GPCR', 'Cytokine–cytokine receptor interaction', 'Allograft Rejection', and 'Graft-versus-host disease'. The majority of enriched GO terms were related to signaling processes, such as 'cAMP-mediated signaling' and 'serotonin receptor signaling pathway' or to the immune system, such as 'acute inflammatory response' and 'regulation of immune system process'. The majority of enriched disease ontologies were related to the immune system, such as 'Autoimmune Diseases', 'Immune System Disease', 'Inflammation', 'Asthma', 'Rheumatic Diseases', 'Dermatitis', 'Celiac Diseases', and 'Organ Transplantation', as well as 'Pregnancy', 'Pregnancy, First Trimester', 'Infertility', 'Habitual Abortion', 'Chorioamnionitis', 'Pre-Eclampsia', and 'Preterm Birth', consistent with observations that women with systemic autoimmune diseases have an elevated risk of delivering preterm (*Kolstad et al., 2020*).

Seven hundred and seventy-one genes lost endometrial expression in the lineage with BPP ≥0.80 (*Figure 2—source data 3*). These genes were enriched in 48 pathways, 42 biological process GO terms, and 3 disease ontologies at FDR ≤0.10 (*Figure 2*). Enriched pathways included 'immune system', 'pregnancy', 'pregnancy first trimester', 'infertility', 'habitual abortion', 'preeclampsia', and 'preterm birth'. Unlike genes that gained endometrial expression in the human lineage, those that lost endometrial expression were enriched in disease ontologies unrelated to the immune system, but did include 'Preterm Birth', as well as 'Selenocysteine Synthesis' and 'Selenoamino Acid Metabolism', the latter two which have been previously implicated in preterm birth by genome-wide association study (GWAS; *Zhang et al., 2017*). In stark contrast, genes that gained (+) or lost (−) endometrial

## Box 1. Classification of genes into not/expressed categories.

A challenge with quantitative gene expression metrics such as RNA-Seq data is defining an expression level that corresponds to functionally active (expressed) genes. Previous studies, however, have shown that an empirically informed operational criterion based on transcript abundance distributions reasonably approximate gene expression categories (*Hebenstreit et al., 2011*; *Kin et al., 2015*; *Wagner et al., 2013*; *Wagner et al., 2012*). *Hebenstreit et al., 2011*, for example, showed that genes can be separated into two distinct groups based on their expression levels: the majority of genes follow a normal distribution and are associated with active chromatin marks at their promoters and thus are likely actively expressed, whereas the remaining genes form a shoulder to the left of this main distribution and are unlikely to be actively expressed. Similarly, *Wagner et al., 2013*; *Wagner et al., 2012* found that gene expression data could be modeled as a mixture of two distributions corresponding to inactive and actively transcribed genes. Based on this mixture model, they proposed an operational criterion for classifying genes into expressed and nonexpressed sets: genes with transcripts per million (TPM) ≥2–4 are likely to be actively transcribed, while genes with TPM <2 are unlikely to be actively transcribed. Furthermore, *Wagner et al., 2013* suggest that the expression cutoff should be chosen depending on the goal of the study. If it is important to reduce false positives (classifying genes as expressed when they are not), then a conservative criterion of TPM ≥4 could be used. In contrast, if it is more important to reduce false-negative gene expression calls (classifying genes as not expressed when they are), then a liberal criterion such as TPM ≥1 could be used. Both *Hebenstreit et al., 2011* and *Wagner et al., 2013* suggest that genes with TPM ~2 are likely to be actively transcribed.

We found that gene expression data ($Log_2$ TPM) from human decidual stromal cells (DSCs) generally followed a normal distribution with a distinct shoulder to the left of the main distribution, which could be modeled as a mixture of two Gaussian distributions with means of TPM ~0.11 and TPM ~19 (*Box 1 – figure 1A*). An empirical cumulative distribution fit (ECDF) to the Gaussian mixture model suggests that genes with TPM = 0.01–1 have less than 50 % probability of active expressin, whereas genes with TPM ≥2 have greater than 75 % probability of active expression (*Box 1—figure 1A*). Next, we grouped genes into three categories, TPM = 0, TPM = 0.01–1, and TPM ≥2, and explored the correlation between these categories and histone marks that are associated with active promoters (H3K4me3) and enhancers (H3K27ac), regions of open chromatin (DNaseI- and FAIRE-Seq), and regions of active transcription (RNA polymerase binding to gene). We found that genes with TPM = 0.01–2 and TPM = 0 were nearly indistinguishable with respect to H3K4me3 marked promoters, H3K27ac marked enhancers, regions of open chromatin assessed by FAIRE-Seq (but not DNaseI-Seq), and most importantly, regions of active transcription as assessed by RNA polymerase binding to gene bodies (*Box 1—figure 1B*). The promoters of genes with TPM ≥2 were also more enriched for binding sites for the progesterone receptor (PGR) and its cofactor GATA2 than genes with TPM <1 (*Box 1 – figure 1C*). These data suggest that genes with TPM <1 are unlikely to be actively expressed while genes with TPM ≥2 have hallmarks of active expression. Therefore, we used the TPM ≥2.0 cutoff to define a gene as expressed. We note, however, that other cutoffs could be used that either increase or decrease the probability that genes are actively expressed.

expression during pregnancy in the stem lineage of primates (+63/−34) did not include terms related to the immune system or pregnancy. Thus, genes that gained or lost endometrial expression in the human lineage are uniquely related to immune regulatory process, autoimmunity, inflammation, and allograft rejection, signaling processes such as cAMP-mediated and serotonin receptor signaling, and well as adverse pregnancy outcomes.

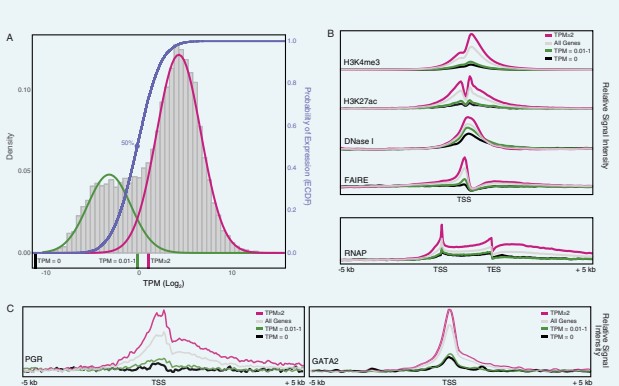

**Box 1—figure 1.** Gene expression and functional genomics data suggest that genes with transcripts per million (TPM) ≥2 are actively expressed.

(A) Distribution of gene expression levels from human decidual stromal cell (DSC) RNA-Seq data. Gray, kernel density estimates of gene expression levels as transcripts per million (TPM) for human RefSeq genes (genes with TPM = 0 are not shown). Expectation-maximization-based Gaussian mixture curve fits of expression data are shown in green and magenta. Empirical cumulative distribution fit (ECDF) to the Gaussian mixture model is shown in blue. Regions of the kernel density plot corresponding to TPM = 0, TPM = 0.01-1, and TPM ≥ 2 are indicated below the plot as black, green, and magenta pink bars, respectively. The point of the ECDF corresponding to a 50% probability of expression is indicated with a blue circle.(B) Correlation of gene expression categories (TPM = 0, TPM = 0.01-1, and TPM ≥ 2) with histone marks that characterize active promoters (H3K4me3), enhancers (H3K27ac), regions of open chromatin (DNaseI and FAIRE), and active transcription (RNAP binding to gene bodies). TSS, transcription start site. TES, transcription end site.(C) Correlation of gene expression categories (TPM = 0, TPM = 0.01-1, and TPM ≥ 2) with progesterone receptor (PGR) and the PGR co-factor GATA2 binding sites. TSS, transcription start site.

## Human recruited genes predominantly remodeled the transcriptome of endometrial stromal cells

The maternal–fetal interface is composed of numerous maternal and fetal cell types including endometrial stromal lineage cells (perivascular, EFSs, and DSCs), uterine natural killer cells (uNKs), decidual macrophage (uMP), dendritic cells (DCs), T helper cells (Th cells), regulatory T cells (Tregs), various innate lymphoid cells (ILCs), and multiple trophoblast cell types (*Suryawanshi et al., 2018*; *Vento-Tormo et al., 2018*; *Wang et al., 2020*). To infer if genes recruited into endometrial expression in the human lineage are enriched in specific cell types, we used a previously published single-cell RNA-Seq (scRNA-Seq) dataset generated from the first trimester human decidua (*Vento-Tormo et al., 2018*) to identify cell types at the maternal–fetal interface (*Figure 3A*; *Figure 3—figure supplement 1*). Next, we determined the observed fraction of human recruited genes expressed in each cell type compared to the expected fraction and used a two-way Fisher exact test to identify cell types that were significantly enriched in human recruited genes. Remarkably, human recruited genes were enriched in five of six endometrial stromal lineage cells, including perivascular endometrial mesenchymal stem cells (pvEMSCs) and four populations of DSCs, as well as plasmocytes, endothelial cells (ECs), DCs, and extravillus cytotrophoblasts (*Figure 3B*). Consistent with these findings, the expression of human recruited genes defines distinct cell types at the maternal–fetal interface (*Figure 3—figure supplement 2*).

Our observation that human recruited genes have predominantly remodeled the transcriptome of endometrial stromal lineage cells prompted us to explore the development and gene expression evolution of these cell types in greater detail. Pseudotime single-cell trajectory analysis of endometrial stromal lineage cells identified six distinct populations corresponding to a perivascular mesenchymal stem cell like endometrial stromal population (pvEMSC) population and five populations of DSCs (DSC1–5), as well as cells between pvEMSCs and DSCs that likely represent nondecidualized ESFs and ESFs that have initiated decidualization (*Figure 4A and B*). In addition, ESFs that decidualize branch into two distinct lineages, which we term lineage 1 DSCs (DSC1–DSC3) and lineage 2 DSCs (DSC4

## Box 2. Genomic features of human recruited genes.

The expression of human recruited genes is enriched in 25 tissues at false discovery rate (FDR) <0.05 (**Box 1 – figure 1** inset), suggesting these genes were predominately recruited into endometrial expression from those tissues. The expression of human recruited genes in human gestation week 9–22 decidua followed a normal distribution that could be modeled as a mixture of two Gaussian distributions with means of transcripts per million (TPM) ~2.8 and ~10.5 (**Box 2—figure 1B**), grouping genes into low and high expression sets around TPM ~4.2 (**Box 2 – figure 1B**). An empirical cumulative distribution fit (ECDF) to the gene expression data also suggests a cutoff at TPM ~4.2, which defines an expression level at which 50 % of genes are binned into either the high or low expression sets. The promoters of human recruited genes with TPM <4.2 and ≥4.2 were indistinguishable with respect to H3K4me3 and H3K27ac signal at promoters and enhancers, DNaseI hypersensitive sites, progesterone receptor (PGR), and GATA2 binding, and RNA polymerase binding to gene bodies; both expression sets were generally enriched in these signals compared to genes with TPM = 0 or random genomic locations (**Box 2 – figure 1C**). In contrast, the promoters of human recruited genes with TPM ≥4.2 are in regions of chromatin with greater nucleosome depletion than recruited genes with TPM <4.2 as assessed by FAIRE-Seq. This observation is consistent with previous studies which found the promoters of highly transcribed genes are preferentially isolated by FAIRE-Seq (**Giresi et al., 2007**; **Nagy et al., 2003**).

A particularly noteworthy human recruited gene is *PRL*, which we previously showed evolved endometrial expression in primates (**Emera et al., 2012a**) and is the most highly expressed human recruited gene in our dataset (**Box 2 – figure 1B**). Remarkably, *PRL* gene has independently coopted transposable elements (TEs) into decidual promoters in multiple Eutherian lineages (**Emera et al., 2012a**; **Gerlo et al., 2006**; **Lynch et al., 2015**), suggesting that TE cooption into decidual promoters may be a widespread phenomenon. Consistent with this hypothesis, human recruited genes with TEs in their promoters and 5'-UTRs had greater H3K4me3 signal and nucleosome depletion as assessed by FAIRE- and DNaseI-Seq (**Box 2 – figure 1D**). The majority of TEs within the promoters and 5'-UTRs of (40%) were primate specific, suggesting they may have played a role in recruiting these genes into endometrial expression (**Box 2 – figure 1E**).

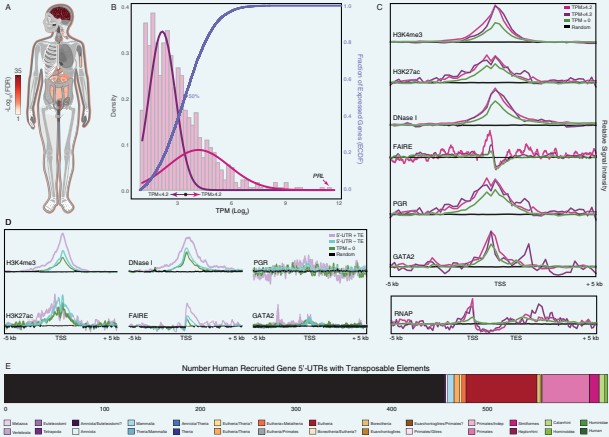

**Box 2—figure 1.** Genomic features of human recruited genes.

(A) Anatogram heatmap showing organs in which the expression of human recruited genes is enriched (the top 15 organs FDR < 0.05).(B) Distribution of human recruited gene expression levels from human gestation week 9-22 decidua RNA-Seq data. Light pink, kernel density estimates of gene expression

levels as transcripts per million (TPM) for human RefSeq genes (genes with TPM < 2 are classified as "not expressed" and are not shown). Expectation-maximization-based Gaussian mixture curve fits of expression data are shown in purple (low expressed genes) and magenta (high expressed genes), the TPM 4.2 cutoff for defining low and high expressed genes is shown as a black circle. Empirical cumulative distribution fit (ECDF) to the gene expression data is shown in blue, the point of the ECDF at which 50% of genes are binned into either the high or low expression sets is indicated with a blue circle.(C) Correlation of gene expression categories (random genomic locations, TPM = 0, TPM < 4.2, and ≥ 4.2) with histone marks that characterize active promoters (H3K4me3), enhancers (H3K27ac), regions of open chromatin (DNaseI and FAIRE), PGR and GATA2 binding sites, and RNAP binding to gene bodies (active transcription). TSS, transcription start site. TES, transcription end site.(D) Correlation of gene expression categories (random genomic locations, TPM = 0, human recruited genes with (+) and without (–) TEs in their promoters) with H3K4me3, H3K27ac, DNaseI, FAIRE, PGR, GATA2, RNAP signal intensities. TSS, transcription start site. TES, transcription end site.(E) Number of transposable element families within the promoters and 5'-UTRs of human recruited genes. Transposable elements are colored by their lineage specificity.

and DSC5) (*Figure 4A and B*). These cell populations differentially express human recruited genes (*Figure 4C*), which are dynamically expressed during differentiation of perivascular cells (PVCs) into lineage 1 and 2 DSCs (*Figure 4D*).

## Co-option of serotonin signaling in human endometrial cells

Genes that were recruited into endometrial expression in the human lineage are enriched the serotonin signaling pathway (*Figure 2*), but a role for serotonin signaling in the endometrium has not

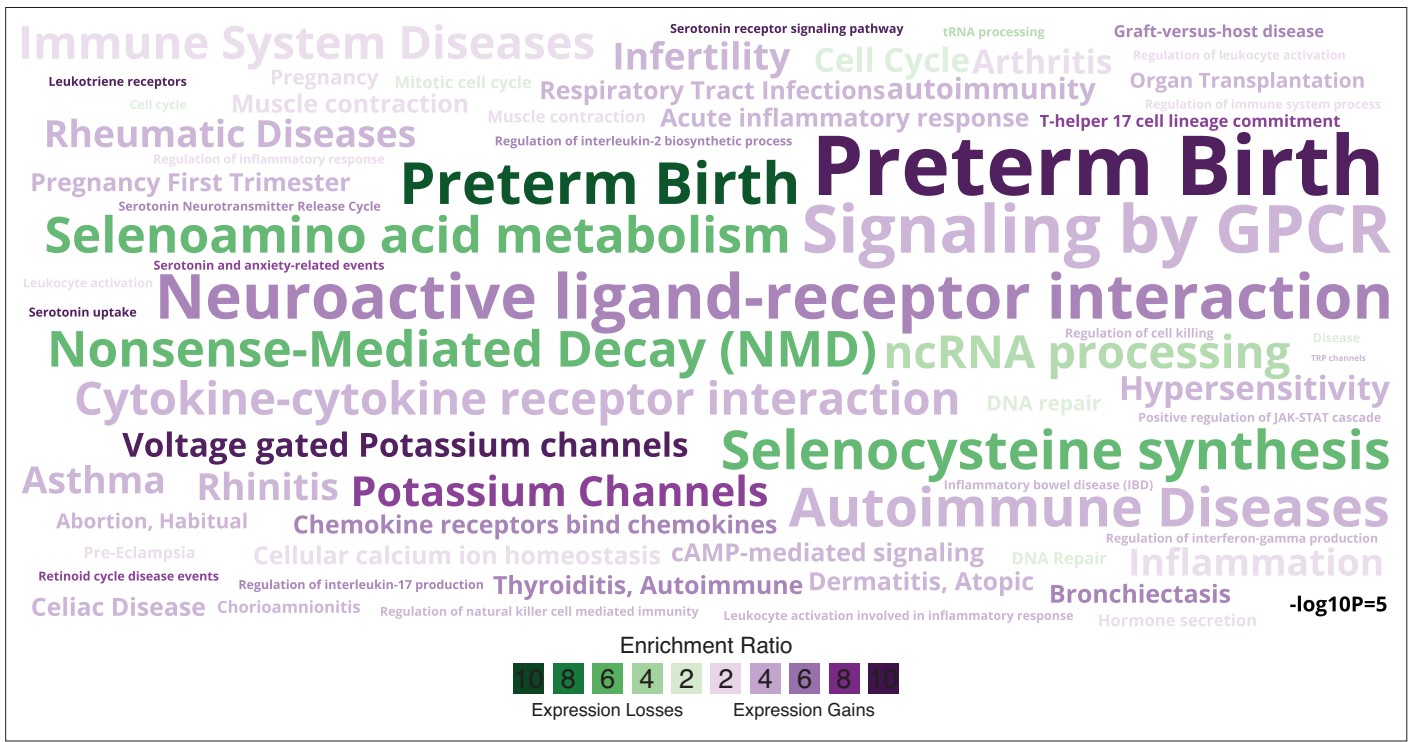

**Figure 2.** Enriched pathways, gene ontologies, and disease ontologies among genes that gained or lost endometrial expression in the Hominoid (human) lineage. Data shown as a WordCloud, with term size proportional to −log10 hypergeometric p value (see inset scale) and colored according to enrichment ratio for genes that gained (purple) or lost (green) endometrial expression.

The online version of this article includes the following figure supplement(s) for figure 2:

**Source data 1.** Custom gmt file used for enrichment tests related to preterm birth.

**Source data 2.** Genes that gained expression (Bayesian posterior probability [BPP] ≥0.80) in the Hominoid (human) lineage.

**Source data 3.** Genes that lost expression (Bayesian posterior probability [BPP] ≥0.80) in the Hominoid (human) lineage.

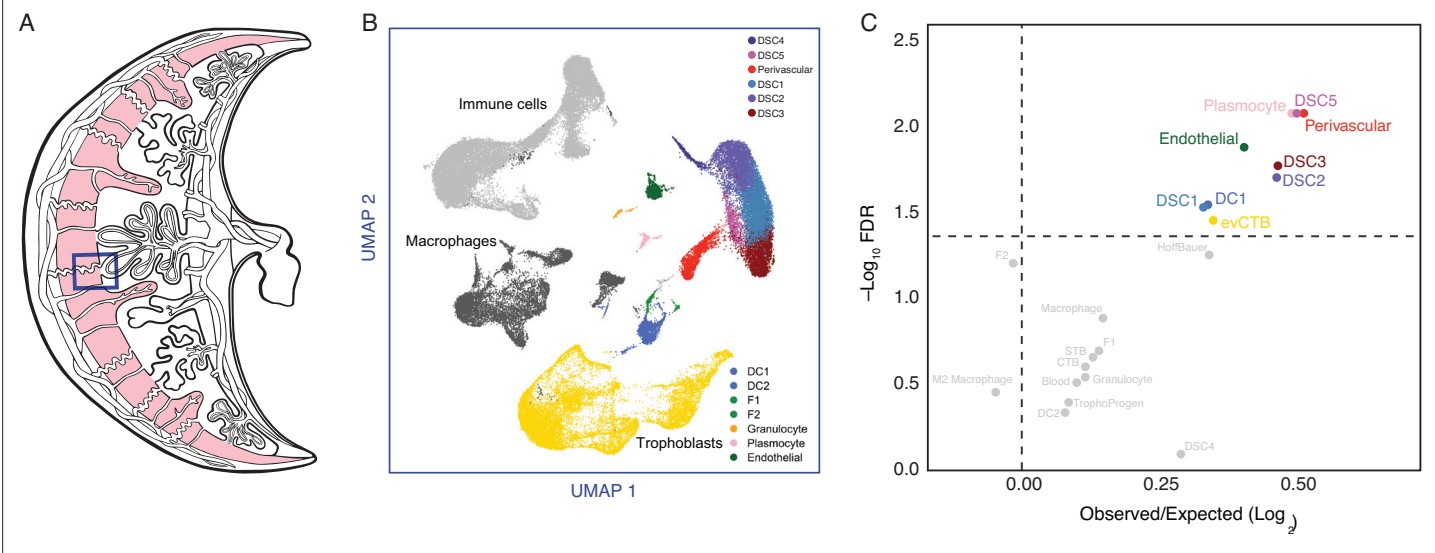

**Figure 3.** The expression of Hominoid (human) recruited genes enriched in endometrial stromal lineage cells. (**A**) Anatogram of the human maternal–fetal interface. The decidua is shown in light pink, single-cell RNA-Seq (scRNA-Seq) data (***Vento-Tormo et al., 2018***) were generated from the region boxed in blue. (**B**) Uniform Manifold Approximation and Projection (UMAP) clustering of cells from the first trimester maternal–fetal interface. Major cell types and lineages are colored. (**C**) Volcano plot showing cell types at the maternal–fetal interface in which Hominoid (human) recruited genes are significantly (false discovery rate [FDR] corrected two-way Fisher's exact test) enriched (Log$_2$ Observed/Expected). Cell types in which recruited genes are significantly enriched (FDR ≤0.05) are labeled and colored as in panel A.

The online version of this article includes the following figure supplement(s) for figure 3:

**Figure supplement 1.** Identification of cell-type populations are the first trimester human maternal-fetal interface.

**Figure supplement 2.** Expression of recruited genes in cell-type populations are the first trimester human maternal-fetal interface.

previously been reported. Among the recruited genes in this pathway is the serotonin receptor *HTR2B*. To explore the history of *HTR2B* expression in the endometrium in greater detail, we plotted extant and ancestral gene expression probabilities on tetrapod phylogeny and found that it independently evolved endometrial expression at least seven times, including in the human lineage (***Figure 5A***). To investigate which cell types express *HTR2B*, we used the scRNA-Seq dataset from the first trimester maternal–fetal interface and found that *HTR2B* expression was almost entirely restricted to the DSC cluster (***Figure 5B***). We further explored the expression dynamics *HTR2B* during decidualization using a scRNA-Seq time-course dataset (***Lucas et al., 2020***) and found that its expression is transiently downregulated during the initial inflammatory decidual phase but upregulated upon the emergence of decidual cells and senescence decidual cells after 4 days of differentiation (***Figure 5—figure supplement 1***). *HTR2B* was also the only serotonin receptor expressed in either human ESFs or DSCs at TPM ≥2 (***Figure 5B*** and ***Figure 5—figure supplement 2A***) and was highly expressed in uterine tissues (***Figure 5—figure supplement 2B***). Additionally, we found that *HTR2B* was only expressed in human and mouse ESFs, but not in ESFs at TPM ≥2 from other species in a previously generated multispecies ESF RNA-Seq dataset (***Figure 5C***).

Pseudotime single-cell trajectory analysis of endometrial stromal lineage cells indicates that *HTR2B* is expressed in most lineage 1 DSCs, which coexpress other genes such as *IL15*, *INSR*, and *PRDM1* (***Figure 5D***); *HTR2B* is also expressed by a minority of lineage 2 DSCs, ESFs, and PVCs (***Figure 5D***). One hundred and ninety-four genes were differentially expressed between *HTR2B*+ and *HTR2B*− DSCs (***Figure 5E***). These genes were enriched in numerous pathways including 'Regulation of Insulin-like Growth Factor (IGF) transport and uptake by Insulin-like Growth Factor Binding Proteins (IGFBPs)', 'Complement and coagulation cascades', 'BMP2–WNT4–FOXO1 Pathway in Human Primary Endometrial Stromal Cell Differentiation', 'IL-18 signaling pathway', and disease ontologies including 'Small-for-dates baby', 'Premature Birth', 'Inflammation', 'Fetal Growth Retardation', 'Pregnancy Complications', 'Hematologic Complications', and 'Spontaneous abortion' (***Figure 5F*** and ***Figure 5—source data 1***).

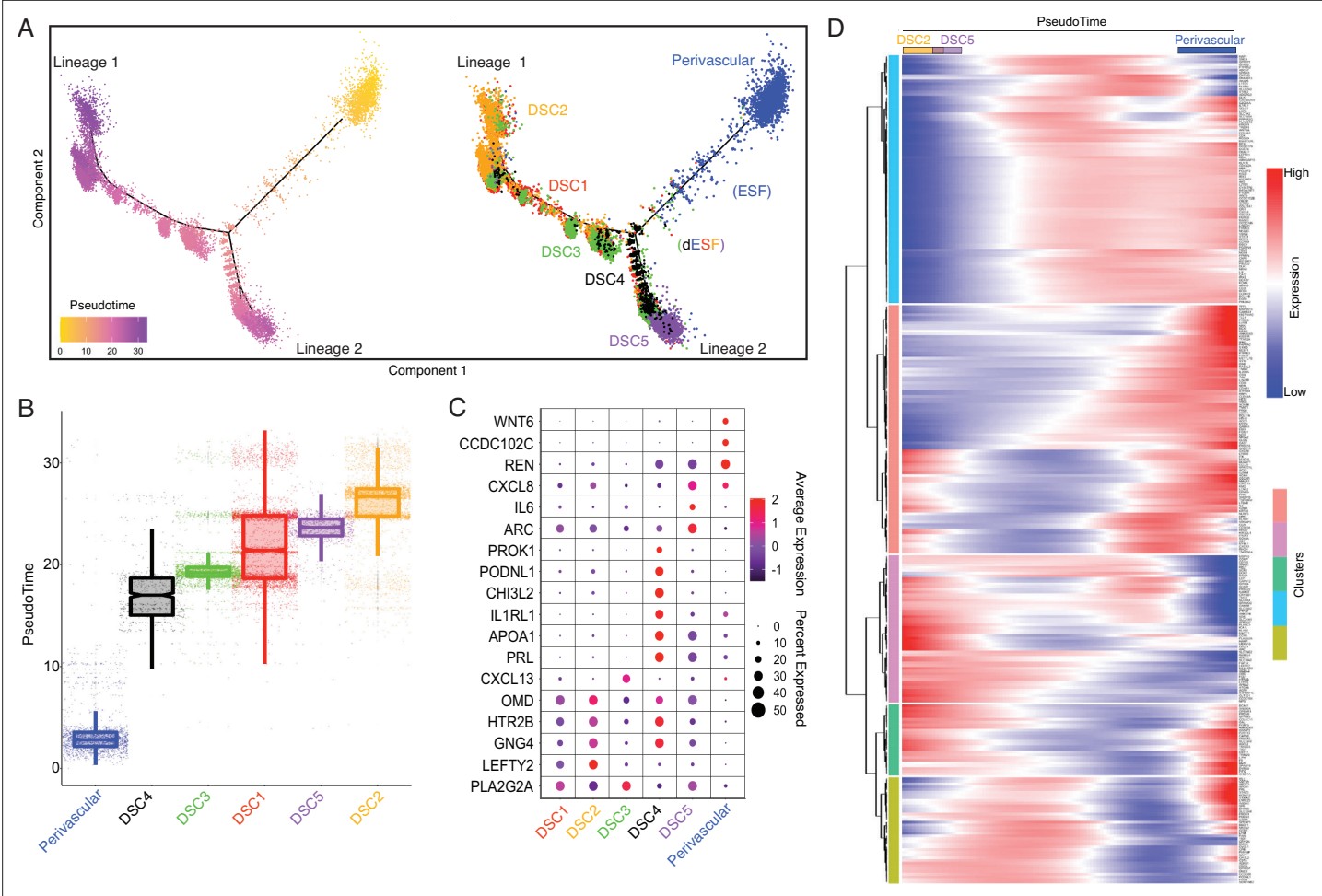

**Figure 4.** Human-gain genes' dynamic expression marks the distinct lineages of decidual cells in the spatiotemporal niche. (**A**) Pseudotime trajectory of endometrial stromal lineages, colored from low (gold) to high (purple) pseudotime (left). Endometrial stromal lineage cell types clustered using the top 2000 differentially expressed genes and projected into a two-dimensional space (right). (**B**) Jittered boxplot illustrating the pseudotime of each cell from A. (**C**) DotPlot illustrating the intensity and abundance of selected human-gain transcript expression between endometrial stromal lineage cell types. Colors represent an average Log$_2$ expression level scaled to the number of unique molecular identification (UMI) values in single cells. The color scale is from blue to red, corresponding to lower to higher expression, respectively. Dot size is proportional to the percent of cells expressing that gene. Genes were selected based on their differential expression on the pseudotime trajectory shown in the previous figure (Benjamini and Hochberg adjusted p value <2.2e−16, Wald test). (**D**) Heatmap showing the kinetics of highly expressed (Log$_2$ scaled average expression >0.5) human-gain genes changing gradually over the trajectory of endometrial stromal lineage cell types shown in panel A. Genes (row) are clustered, and cells (column) are ordered according to the pseudotime progression.

To determine if *HTR2B* expression was regulated by progesterone, we used previously published RNA-Seq data from human ESFs and ESFs differentiated into DSCs with cAMP/progesterone (*Mazur et al., 2015*). *HTR2B* was highly expressed in ESFs and downregulated during differentiation (decidualization) by cAMP/progesterone into DSCs (*Figure 6A* and *Figure 6—figure supplement 1*). *HTR2B* has hallmarks of an expressed gene in DSCs, including residing in a region open chromatin assessed by previously published FAIRE-Seq data (*Figure 6B*), an H3K4me3 and H3K27ac marked promoter and polymerase II binding, as well as a promoter that makes long-range loops to binding sites for transcription factors that orchestrate decidualization such as the progesterone receptor A isoform (PGR-A), FOXO1, FOSL2, GATA2, and NR2F2 (COUP-TFII) in previously published ChIP-Seq data (see methods) (*Figure 6B*). The *HTR2B* promoter also makes several long-range interactions to transcription factor-bound sites as assessed by H3K27ac HiChIP data generated from a normal hTERT-immortalized endometrial cell line (E6E7hTERT; see methods) (*Figure 6B*). Consistent with regulation by these transcription factors, knockdown of *PGR*, *FOXO1*, and *GATA2* upregulated *HTR2B* in DSCs (*Figure 6C*). *HTR2B* is also differentially regulated throughout menstrual cycle (*Figure 6D*) and

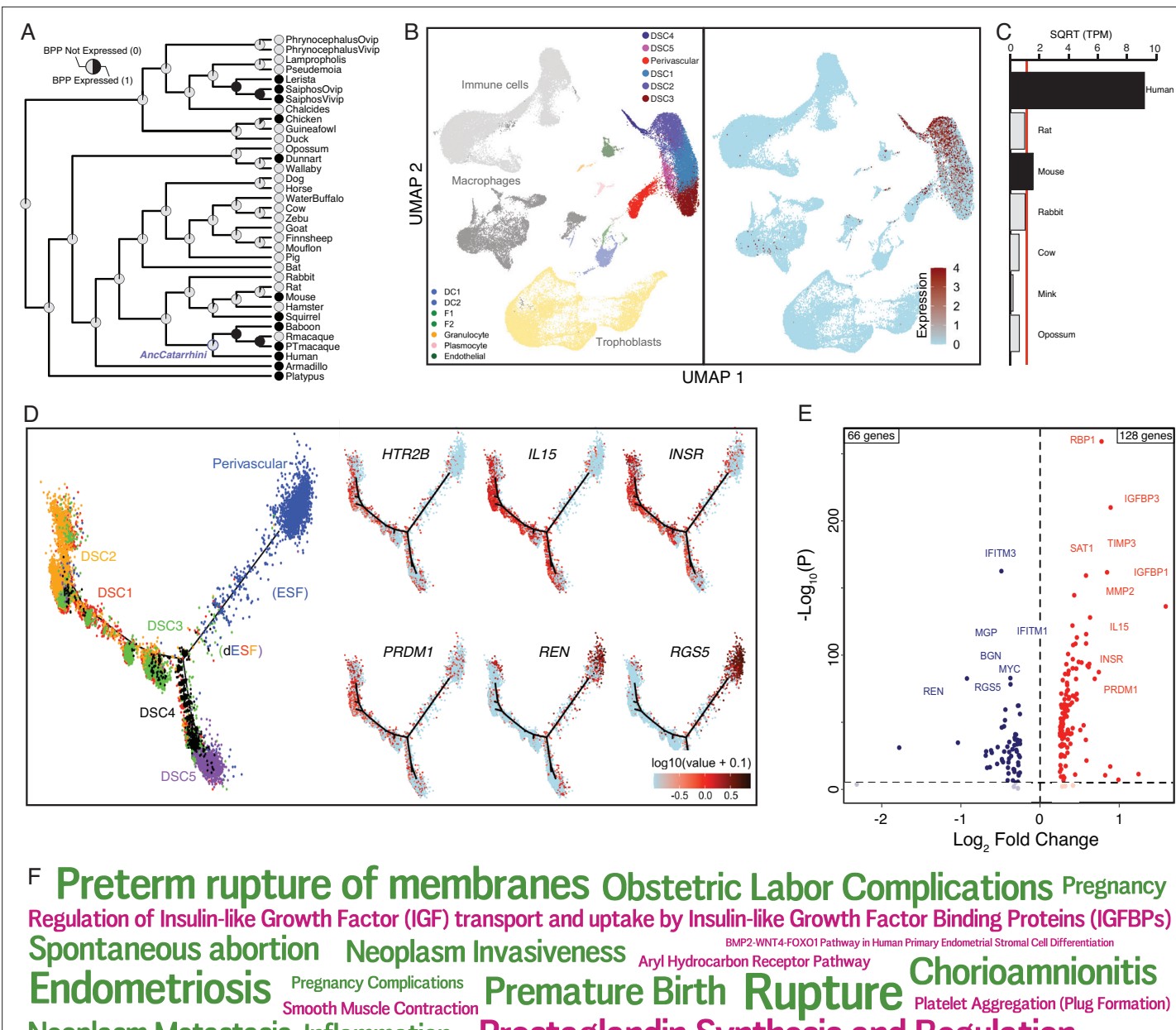

**Figure 5.** The serotonin receptor *HTR2B* evolved to be expressed in decidual stromal cells at the maternal–fetal interface. (**A**) Ancestral construction of *HTR2B* expression in gravid/pregnant endometrium. Pie charts indicate the Bayesian posterior probability (BPP) that *HTR2B* is expressed (state 1) or not expressed (state 0). (**B**) UMAP clustering of cells from the first trimester maternal–fetal interface, decidual stromal cell (DSC) clusters are labeled and highlighted (left). Feature plot based on the UMAP plot showing the single-cell expression of *HTR2B* in the endometrial stromal lineage cells. (**C**) Average expression of *HTR2B* in RNA-Seq data from human, rat, mouse, rabbit, cow, mink, and opossum endometrial stromal fibroblasts (ESFs). Data are shown as square root (SQRT) transformed transcripts per million (TPM), n = 2. (**D**) Pseudotime trajectory of endometrial stromal fibroblast lineage cells. Monocle2 visualization of five distinct clusters of DSCs and perivascular trajectories using the top 2000 differentially expressed genes projected into a two-dimensional space. *HTR2B*, *IL15*, *INSR*, *PRDM1*, *REN*, and *RGS5* expression (log-transformed counts) in individual cells are shown in red along the pseudotime trajectory. *IL15*, *INSR*, and *PRDM1* mark DSCs, *REN* and *RGS5* mark perivascular and decidualizing ESFs (dESFs). (**E**) Volcano plot showing genes that are differentially expressed between *HTR2B+ and HTRB−* decidual stromal cells. Horizontal dashed line indicates −Log$_{10}$+ = 2 (FDR corrected two-way Fisher's exact test). (**F**) Word Cloud showing enriched pathways (pink) and disease ontologies (green) in which genes that are differentially expressed between HTR2B+ and HTR2B− cells are enriched.

The online version of this article includes the following figure supplement(s) for figure 5:

*Figure 5 continued on next page*

**Figure 5 continued**

**Source data 1.** Genes that are differentially expressed between *HTR2B*⁺ and *HTR2B*⁻ cells, and the pathways/disease ontologies in which they are enriched.

**Figure supplement 1.** Additional single-cell RNA-Seq [scRNA-Seq] analyses of *HTR2B*, *PDCD1LG2*, and *CORIN* across multiple endometrial datasets.

**Figure supplement 2.** HTR2B is the only serotonin receptor expressed endometrial cells.

pregnancy (*Figure 6E*), and is expressed in DSCs in the endometrium during the window of implantation (*Figure 6—figure supplement 2*).

To test if human ESFs and DSCs were responsive to serotonin, we transiently transfected each cell type with reporter vectors that drive luciferase expression in response to activation the AP1 (Ap1_pGL3-Basic[minP]), MAPK/ERK (SRE_pGL3-Basic[minP]), RhoA GTPase (SRF_pGL3-Basic[minP]), and cAMP/PKA (CRE_pGL3-Basic[minP]) signaling pathways, and used a Dual Luciferase Reporter assay to quantify luminescence 6 hr after treatment with either 0, 50, 200, or 1000 μM serotonin. Two pathway reporters were responsive to serotonin: (1) the serum response element (SRE) reporter in DSCs treated with 1000 μM serotonin (unpaired mean difference between is 1.35 [95.0% CI 0.624, 2.69], two-sided permutation *t*-test p = 0.00); and (2) the cAMP/PKA response element (CRE) reporter in ESFs treated with 1000 μM serotonin (unpaired mean difference between is 0.296 [95.0% CI 0.161, 0.43], two-sided permutation *t*-test p = 0.00) and in DSCs treated with 50 μM (unpaired mean difference = −10.1 [95% CI −13.8, −6.28], two-sided permutation *t*-test p = 0.001), 200 μM (unpaired mean difference = 17.4 [95% CI 11.6, 24.6], two-sided permutation *t*-test p = 0.0004), and 1000 μM serotonin (unpaired mean difference is 16.7 [95 %CI 7.67, 26.8], two-sided permutation *t*-test p = 0.006) (*Figure 6F* and *Figure 6—figure supplement 3*).

## Co-option of *PDCD1LG2* (PD-L2) in human endometrial cells

Human recruited genes are enriched numerous immune pathway (*Figure 2*), among these genes are the PD-1 ligand *PDCD1LG2* (PD-L2) (*Figure 7A*). We found that *PDCD1LG2* was expressed by several cell types at the first trimester maternal–fetal interface, including DCs, macrophages, ESFs and DSCs, and multiple trophoblast lineages (*Figure 7B*), and is highly expressed in uterine tissues (*Figure 7—figure supplement 1*). While each of these cell-type populations has individual cells with high-level *PDCD1LG2* expression, only 3%–5% of DSCs, 3 % of DCs, 14 % of macrophage, and 66 % of cytotrophoblasts express *PDCD1LG2* (*Figure 7C*). Consistent with recent recruitment in the human lineage, *PDCD1LG2* was highly expressed in human but either moderately or not expressed in ESFs from other species (*Figure 7D*; *Figure 5—figure supplement 1*). The human *PDCD1LG2* locus has the hallmarks of an actively expressed gene, such as a promoter marked by H3K27ac, H3K4me3, and H3K4me1, and binding sites for several transcription factors in previously published ChIP-Seq data from DSCs (*Figure 7E*). The *PDCD1LG2* promoter also makes several long-range interactions to transcription factor-bound sites, including downstream site that is in the region of open chromatin and bound by PGR/GATA/FOXO1 (*Figure 7E*). *PDCD1LG2* was highly expressed in ESFs and DSCs (*Figure 7F*) but downregulated by cAMP/progesterone treatment (*Figure 7G*). Knockdown of *PGR* and *FOXO1* up- and downregulated *PDCD1LG2* in DSCs, respectively (*Figure 7G*). *PDCD1LG2* introns also contain several single nucleotide polymorphisms (SNPs) previously associated with gestational duration and number of lifetime pregnancies as assessed by GWAS (*Aschebrook-Kilfoy et al., 2015*; *Sakabe et al., 2020*; *Zhang et al., 2017*), albeit with marginal p values, implicating *PDCD1LG2* in regulating gestation length (*Figure 7E*).

## Co-option of *CORIN* into human endometrial cells

Among the human recruited genes enriched in disease ontologies related to preeclampsia (*Figure 2*) is *CORIN* (*Figure 8A*), a serine protease which promotes uterine spiral artery remodeling and trophoblast invasion (*Cui et al., 2012*; *Yan et al., 2000*). We found that *CORIN* was exclusively expressed by a subset of endometrial stromal lineage cells (*Figure 8B and C*), dramatically upregulated in DSCs by cAMP/progesterone treatment (*Figure 8D*), and highly expressed in uterine tissues (*Figure 8—figure supplement 1*; *Figure 5—figure supplement 1*). The *CORIN* locus has hallmarks of an actively expressed gene in DSCs, including a promoter in a region of open chromatin assessed by previously published ATAC- and DNase-Seq data and marked by H3K4me3 in previously published ChIP-Seq data

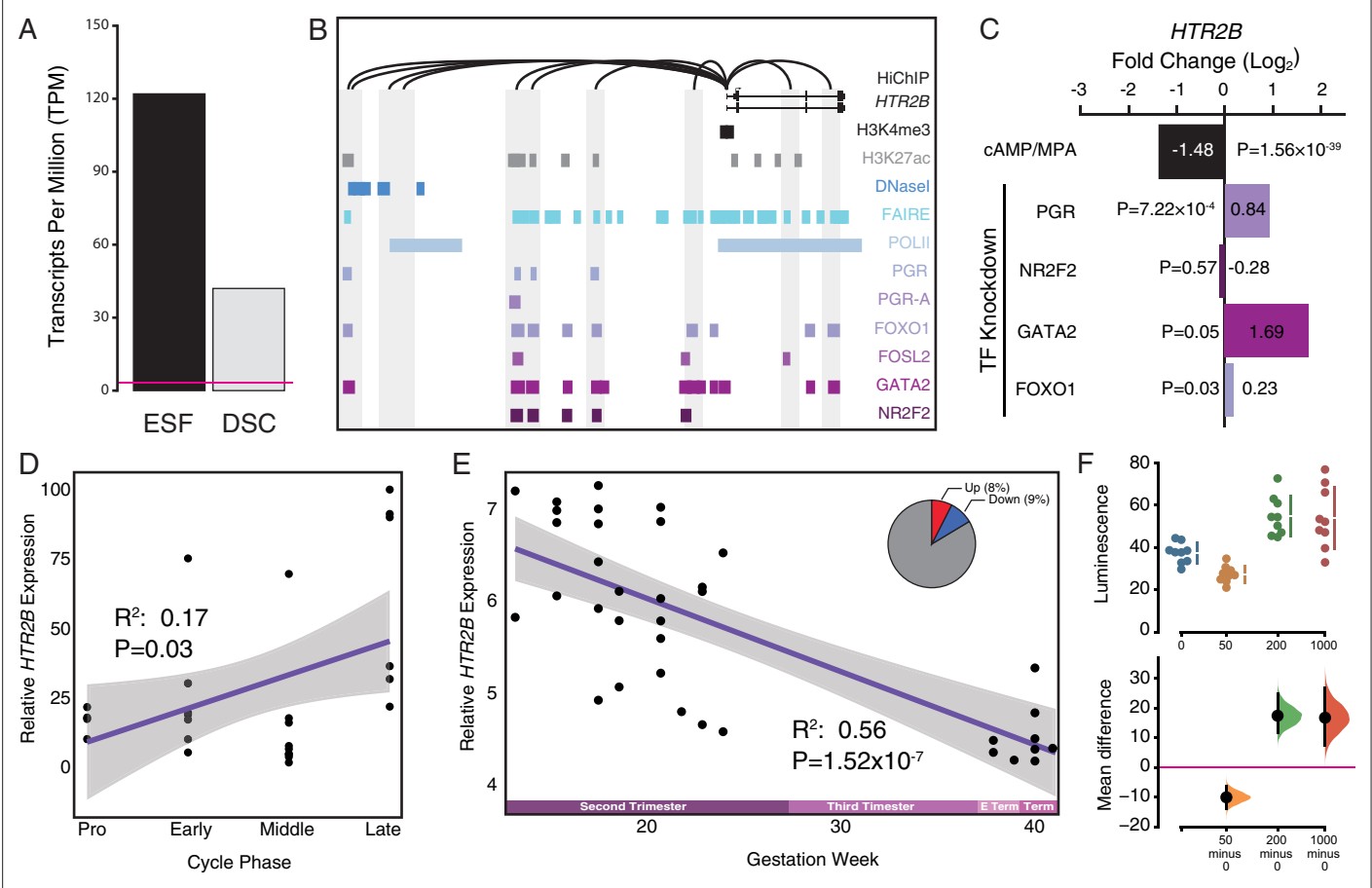

**Figure 6.** Co-option of serotonin signaling in the endometrium. (**A**) *HTR2B* expression in human endometrial stromal fibroblasts (ESFs) is downregulated by cyclic adenosine monophosphate (cAMP)/progesterone treatment for 48 hr (decidualization into decidual stromal cells [DSCs]). Transcript abundance in RNA-Seq data is shown as transcripts per million (TPM). (**B**) Regulatory elements in human DSCs at the *HTR2B* locus. ChIP-Seq peaks shown for H3K4me3, H3K27ac, polymerase II (POLII), progesterone receptor (PGR) and the PGR-A isoform, FOXO1, FOSL2, GATA2, and NR2F2 (COUP-TFII). Regions of open chromatin are shown from DNaseI- and FAIRE-Seq. Chromatin loops inferred from H3K27ac HiChIP are shown as black arcs connecting the *HTR2B* promoter to other locations in the genome shown in gray. (**C**) *HTR2B* expression is downregulated by in vitro decidualization of ESFs into DSC by cAMP/progesterone treatment, and upregulated by small interfering RNA (siRNA)-mediated knockdown of PGR, GATA2, and FOXO1, but not NR2F2. n = 3 per transcription factor knockdown. (**D**) Relative expression of HTR2B in the proliferative (n = 6), early (n = 4), middle (n = 9), and late (n = 8) secretory phases of the menstrual cycle. Note that outliers are excluded from the figure but not the regression; 95% CI is shown in gray. Gene expression data from *Talbi et al., 2006*. (**E**) Relative expression of *HTR2B* in the basal plate from midgestation to term (14–40 weeks, n = 36); 95% confidence interval (CI) is shown in gray. Inset, percent of up- and downregulated genes between weeks 14–19 and 37–40 of pregnancy (false discovery rate [FDR] ≤0.10). Gene expression data from *Winn et al., 2007*. (**F**) Cumming estimation plot showing mean difference in luminescence for the serotonin dose response. Upper axis shows relative luminescence of human decidual stromal cells (hDSCs) transiently transfected with a luciferase expression vector that drives the transcription of the luciferase reporter gene from a cAMP/PKA response element (pGL4.29[luc2P/CRE/Hygro]) 6 hr after treatment with serotonin (50, 200, and 1000 μM) or vehicle control (water). Lower axes, mean differences are plotted as bootstrap sampling distributions (n = 5000; the confidence interval is bias-corrected and accelerated). Each mean difference is depicted as a dot. Each 95 % confidence interval is indicated by the vertical error bars. p values indicate the likelihoods of observing the effect sizes, if the null hypothesis of zero difference is true.

The online version of this article includes the following figure supplement(s) for figure 6:

**Figure supplement 1.** Volcano plot of gene expression changes between endometrial stromal fibroblasts (ESFs) and decidual stromal cells (DSCs).

**Figure supplement 2.** Immunofluorescent staining of HTR2B in human endometria.

**Figure supplement 3.** Pathway activation screen.

(*Figure 8E*). The *CORIN* promoter also makes long-range interactions to transcription factor-bound sites as assessed by HiChIP, including an upstream site bound by PGR, FOSL2, GATA2, FOXO1, and NR2F2 in previously published ChIP-Seq data from DSCs (*Figure 8E*). Consistent with these observations, knockdown of PGR, NR2F2, and GATA2 downregulated *CORIN* expression in DSCs (*Figure 8F*).

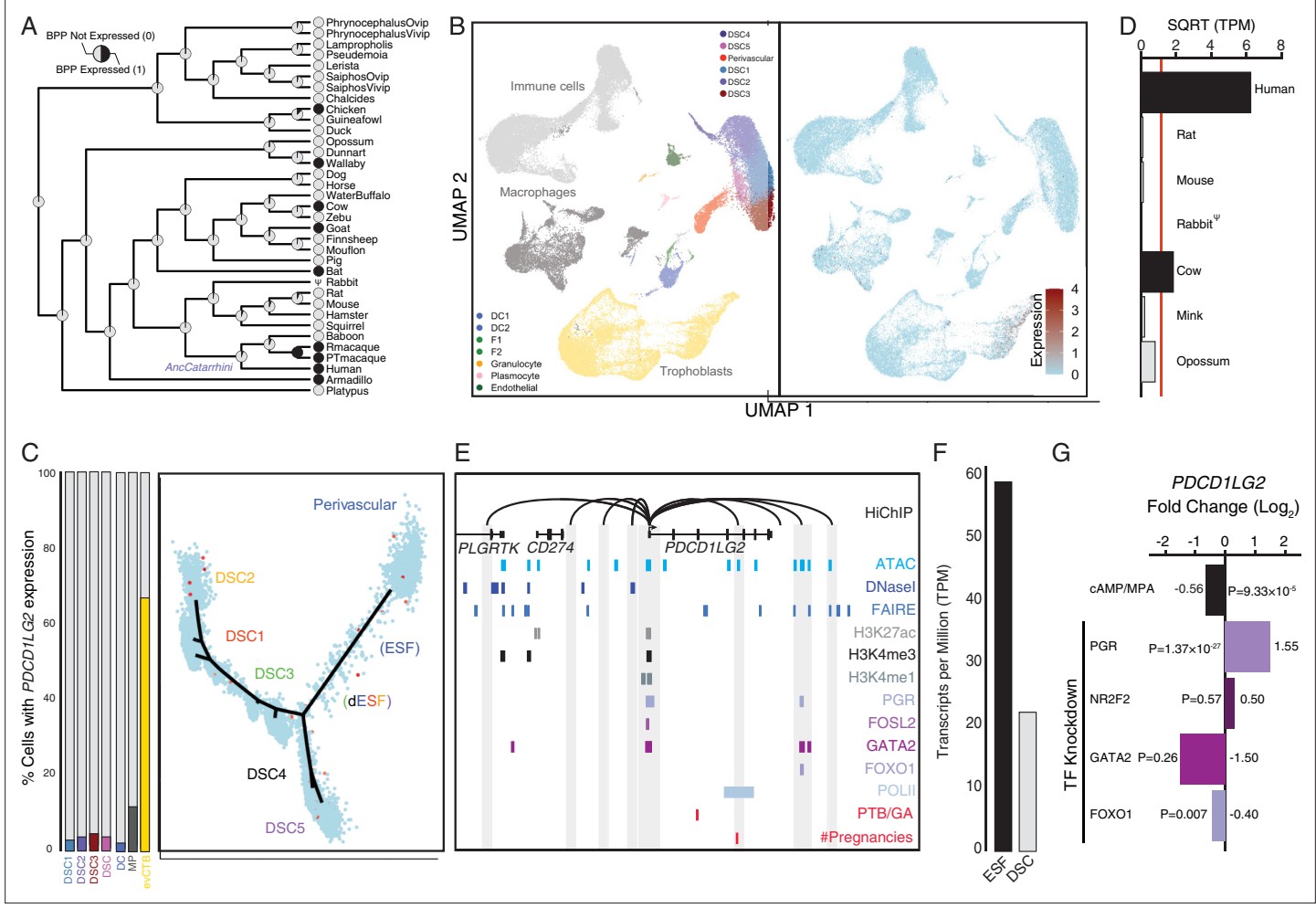

**Figure 7.** Co-option of *PDCD1LG2* into endometrial cells. (**A**) Ancestral construction of *PDCD1LG2* expression in gravid/pregnant endometrium. Pie charts indicate the Bayesian posterior probability (BPP) that *PDCD1LG2* is expressed (state 1) or not expressed (state 0). (**B**) UMAP clustering of cells from the first trimester maternal–fetal interface. *PDCD1LG2* expression (log-transformed counts) in individual cells is shown in red. (**C**) Left: proportion of cell types at the maternal–fetal interface that express *PDCD1LG2*. Only cell types that express *PDCD1LG2* are shown as a 100 % stacked bar chart: decidual stromal cell populations 1–3 (DSC1–3), average expression in DSC1–3, dendritic cells (DCs), macrophage (MP), and extravillus cytotrophoblasts (evCTB). Right: pseudotime trajectory of endometrial stromal fibroblast lineage cells. Monocle2 visualization of five distinct clusters of DSCs and perivascular trajectories projected into a two-dimensional space. *PDCD1LG2* expression (log-transformed counts) in individual cells is shown in red along the pseudotime trajectory. (**D**) Average expression of *PDCD1LG2* in RNA-Seq data from human, rat, mouse, rabbit, cow, mink, and opossum endometrial stromal fibroblasts (ESFs). Data are shown as square root (SQRT) transformed transcripts per million (TPM), *n* = 2. (**E**) Regulatory elements in human DSCs at the *PDCD1LG2* locus. ChIP-Seq peaks shown for H3K4me1, H3K4me3, H3K27ac, polymerase II (POLII), progesterone receptor (PGR), FOXO1, FOSL2, GATA2, and NR2F2 (COUP-TFII). Regions of open chromatin are shown from DNaseI-, ATAC-, and FAIRE-Seq. Chromatin loops inferred from H3K27ac HiChIP are shown as black arcs connecting the *PDCD1LG2* promoter to other locations in the genome shown in gray. The location of SNPs implicated by genome-wide association study (GWAS) in preterm birth is shown in red. (**F**) *PDCD1LG2* expression in human ESFs is downregulated by cyclic adenosine monophosphate (cAMP)/progesterone treatment for 48 hr (decidualization into DSCs). Transcript abundance in RNA-Seq data is shown as TPM. (**G**) *PDCD1LG2* expression is downregulated by in vitro decidualization of ESFs into DSC by cAMP/progesterone treatment and by siRNA-mediated knockdown of FOXO1. siRNA-mediated knockdown of PGR upregulated *PDCD1LG2* expression, while there was no effect after siRNA-mediated knockdown of NR2F2 or GATA2. *n* = 3 per transcription factor knockdown.

The online version of this article includes the following figure supplement(s) for figure 7:

**Figure supplement 1.** Expression of *PDCD1LG2* in GTEx tissues.

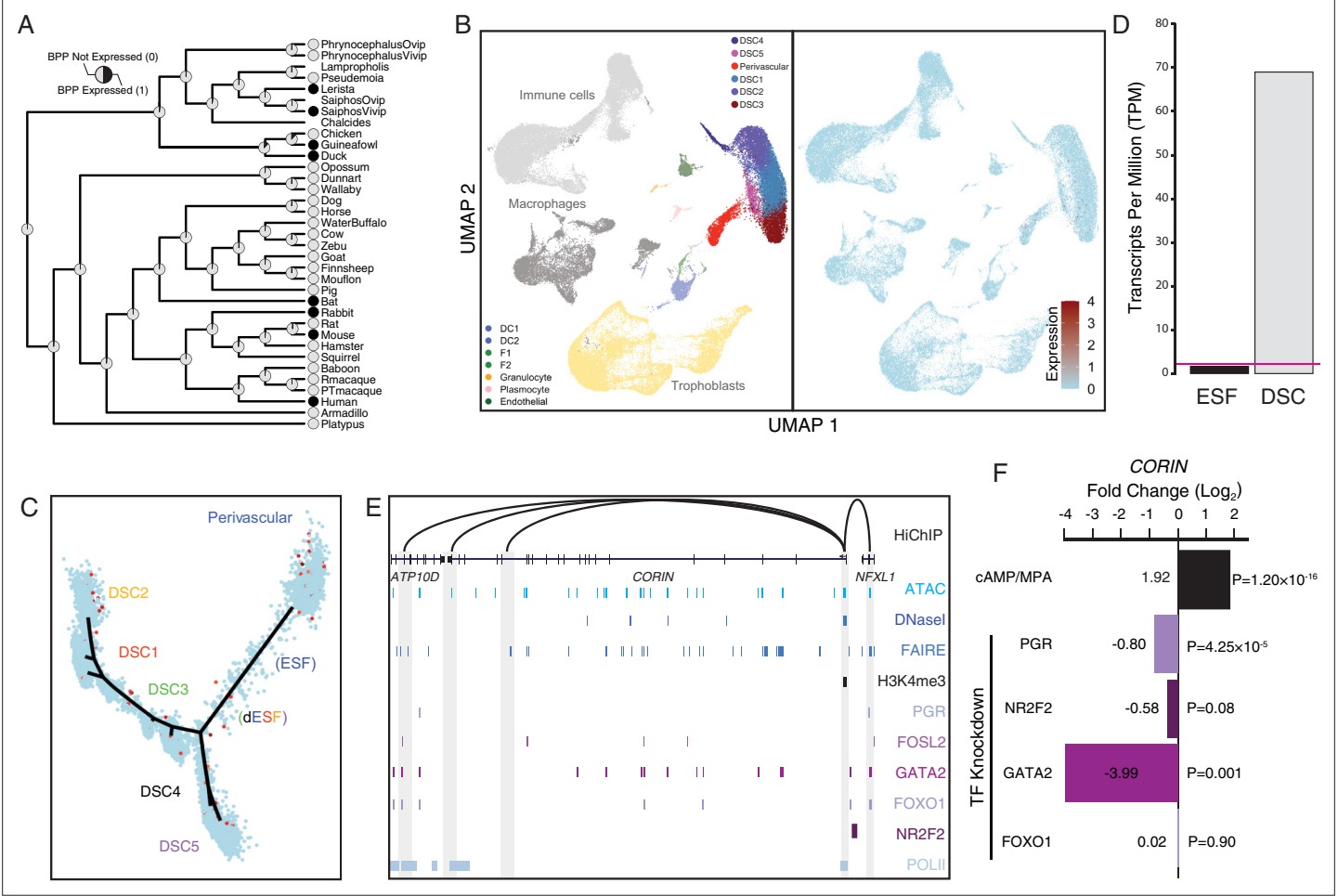

**Figure 8.** Co-option of *CORIN* into endometrial cells. (**A**) Ancestral construction of *CORIN* expression in gravid/pregnant endometrium. Pie charts indicate the Bayesian posterior probability (BPP) that *CORIN* is expressed (state 1) or not expressed (state 0). (**B**) UMAP clustering of cells from the first trimester maternal–fetal interface. *CORIN* expression (log-transformed counts) in individual cells is shown in red. (**C**) Pseudotime trajectory of endometrial stromal fibroblast lineage cells. Monocle2 visualization of five distinct clusters of decidual stromal cells (DSCs) and perivascular trajectories projected into a two-dimensional space. *CORIN* expression (log-transformed counts) in individual cells is shown in red along the pseudotime trajectory. (**D**) *CORIN* expression in human endometrial stromal fibroblasts (ESFs) is upregulated by cAMP/progesterone treatment for 48 hr (decidualization into DSCs). Transcript abundance in RNA-Seq data is shown as transcripts per million (TPM). (**E**) Regulatory elements in human DSCs at the *CORIN* locus. ChIP-Seq peaks shown for H3K4me3, polymerase II (POLII), progesterone receptor (PGR), FOXO1, FOSL2, GATA2, and NR2F2 (COUP-TFII). Regions of open chromatin are shown from DNaseI-, ATAC-, and FAIRE-Seq. Chromatin loops inferred from H3K27ac HiChIP are shown as black arcs connecting the *CORIN* promoter to other locations in the genome shown in gray. (**F**) *CORIN* expression is upregulated by in vitro decidualization of ESFs into DSC by cyclic adenosine monophosphate (cAMP/progesterone treatment, and down)regulated by siRNA-mediated knockdown of PGR and GATA2, but not FOXO1 or NR2F2. $n = 3$ per transcription factor knockdown.

The online version of this article includes the following figure supplement(s) for figure 8:

**Figure supplement 1.** Expression of *CORIN* in GTEx tissues.

## Discussion

Reconstructing the developmental and evolutionary history of anatomical systems is essential for a causally complete explanation for the origins and progression of disease, which has led to the synthesis of evolution and medicine ('evolutionary medicine') (*Benton et al., 2021*). We used comparative transcriptomics to explore how the functions of the maternal side (endometrium) of the maternal–fetal interface evolved, and found that hundreds of genes gained or lost endometrial expression in the human lineage. These recruited genes are enriched in immune functions, signaling processes and genes associated with adverse pregnancy outcomes such as infertility, recurrent spontaneous

abortion, preeclampsia, and preterm birth. Among these genes are those that may contribute to a previously unknown maternal–fetal communication system (*HTR2B*), augment maternal–fetal immunotolerance (*PDCD1LG2* also known as *PD-L2*), and promote vascular remodeling and deep placental invasion (*CORIN*).

## Human-specific remodeling of the endometrial stromal cell transcriptome

The maternal–fetal interface is composed of numerous maternal cell types, all which could have been equally impacted by genes that were recruited into endometrial expression in the human lineage. It is notable, therefore, that the expression of these genes is predominately enriched in endometrial stromal lineage cells, including perivascular mesenchymal stem cells and multiple populations of DSCs. These data suggest that remodeling of the transcriptome and functions of the endometrial stromal cell lineage has played a particularly important role in the evolution of human-specific pregnancy traits. It is also interesting to note that DSCs evolved in the stem lineage of Eutherian mammals (*Carter and Mess, 2017*; *Gellersen et al., 2007*; *Gellersen and Brosens, 2003*; *Kin et al., 2016*; *Kin et al., 2015*; *Mess and Carter, 2006*), coincident with a wave of gene expression recruitments and losses that also dramatically remodeled their transcriptomes (*Kin et al., 2015*; *Lynch et al., 2015*). Thus, the endometrial stromal cell lineage has repeatedly been the target of evolutionary changes related to pregnancy, highlighting the importance of DSCs in the origins and divergence of pregnancy traits. These data also suggest that endometrial stromal lineage cells may play a dominant role in the ontogenesis of adverse pregnancy outcomes.

## Co-option of serotonin signaling in human endometrial cells

Unexpectedly, human recruited genes are enriched in the serotonin signaling pathway, such as the serotonin receptor *HTR2B*. Though a role for serotonin in the endometrium has not previously been reported, we found that serotonin treatment effected RAS/MAPK(ERK) and cAMP/PKA signaling pathways, which are essential for decidualization, and that *HTR2B* is dynamically expressed during menstrual cycle and pregnancy, reaching a low at term. Previous studies have shown that the human placenta is a source of serotonin throughout gestation (*Clark et al., 1980*; *Kliman et al., 2018*; *Laurent et al., 2017*; *Ranzil et al., 2019*; *Rosenfeld, 2020*). Remarkably, a body of early literature suggests serotonin might trigger parturition. For example, levels of both serotonin (5-HT) and 5-hydroxyindoleacetic acid (5-HIAA), the main metabolite of serotonin, are highest in amniotic fluid near term and during labor (*Jones and Pycock, 1978*; *Koren et al., 1961*; *Loose and Paterson, 1966*; *Tu and Wong, 1976*) while placental monoamine oxidase activity (which metabolizes serotonin) is lowest at term (*Koren et al., 1965*). Furthermore, a single dose of the monoamine oxidase inhibitor paraglyline hydrochloride can induce abortion in humans and other animals (*Koren et al., 1966*) Consistent with a potential role in regulating gestation length and parturition, use of selective serotonin reuptake inhibitors is associated with preterm birth (*Eke et al., 2016*; *Grzeskowiak et al., 2012*; *Huybrechts et al., 2014*; *Ross et al., 2013*; *Sujan et al., 2017*; *Yonkers et al., 2012*). 5-HIAA also inhibits RAS/MAPK signaling, potentially by competing with serotonin for binding sites on serotonin receptors (*Chen et al., 2011*; *Klein et al., 2018*; *Schmid et al., 2015*). Collectively, these data suggest a mechanistic connection between serotonin/5-HTAA, and the establishment, maintenance, and cessation of pregnancy.

## Co-option of *PDCD1LG2* (PD-L2) into human endometrial cells

Among the genes with immune regulatory roles that evolved endometrial expression in the human lineage is the programmed cell death protein 1 (PD-1) ligand *PDCD1LG2*. PD-1, a member of the immunoglobulin superfamily expressed on T cells and pro-B cells, regulates a critical immune checkpoint that plays an essential role in downregulating immune responses and promoting self-tolerance by suppressing T-cell inflammatory activity (*Patsoukis et al., 2020*). PD-1 has two ligands, *CD274* (PD-L1) and *PDCD1LG2* (PD-L2), which upon binding PD-1 promote apoptosis in antigen-specific T cells and inhibit apoptosis in anti-inflammatory Tregs (*Patsoukis et al., 2020*). Unlike *CD274*, which is constitutively expressed at low levels in numerous cell types and induced by IFN-gamma, *PDCD1LG2* expression is generally restricted to professional antigen-presenting cells (APCs) such as DCs and macrophages and has a fourfold stronger affinity for PD-1 than does *CD274* (*Ghiotto et al., 2010*;

*Latchman et al., 2001*; *Sharpe et al., 2007*; *Sharpe and Pauken, 2018*) Remarkably, this higher affinity emerged in the Eutherian stem lineage (*Philips et al., 2020*). These data suggest that a subpopulation of human DSCs have co-opted some of the immune regulatory functions of professional APCs, which may have been significantly augmented in the human lineage. While more mechanistic studies will help define the precise role of decidual cells in the establishment and maintenance of maternal–fetal immunotolerance, a role for decidual *PDCD1LG2* in pregnancy is strongly suggested by its association with variants linked to gestational length and number of lifetime pregnancies (parity) (*Aschebrook-Kilfoy et al., 2015*; *Sakabe et al., 2020*; *Zhang et al., 2017*).

## Co-option of *CORIN* into human endometrial cells

Placental invasiveness varies dramatically in Eutherians, but the cellular and molecular mechanisms responsible for this variation are ill defined. One of the genes that may play a role in the evolution of deeply invasive hemochorial placentation is the serine protease *CORIN*, which converts pro-atrial natriuretic peptide (pro-ANP) to biologically active ANP (*Yan et al., 2000*). CORIN-mediated ANP production in the uterus during pregnancy has been shown to promote spiral artery remodeling and trophoblast invasion (*Cui et al., 2012*). These data implicate co-option of *CORIN* into endometrial expression may have contributed to the evolution of particularly deep trophoblast invasion and extensive spiral artery remodeling in humans and other great apes (*Carter et al., 2015*; *Pijnenborg et al., 2011a*; *Pijnenborg et al., 2011b*; *Soares et al., 2018*). *CORIN* expression is also significantly lower in patients with preeclampsia than in normal pregnancies (*Cui et al., 2012*), suggesting that the co-option of *CORIN* into human endometrium may predispose humans to preeclampsia. Additional evolutionary and molecular studies will be required to establish a mechanistic connection between the co-option of *CORIN* into the endometrium, the evolution of hemochorial placentation, and the origins of preeclampsia in the human lineage.

## Caveats and limitations

A limitation of this study is our inability to determine with precise phylogenetic resolution the lineages in which some gene expression changes occurred. For example, we lack pregnant endometrial samples from Hominoids other than humans (chimpanzee/bonobo, gorilla, orangutan, and gibbon/siamang), thus we are unable to identify truly human-specific gene expression changes. Similarly, we lack endometrial gene expression data from multiple human populations exposed to differing environmental stresses, and therefore are unable to determine the range of physiologically 'normal' gene expression or the reaction norms of individual and collective gene expression levels. Our functional genomic and experimental studies are also restricted to an in vitro cell culture system, which makes it difficult to assess the in vivo impact of gene expression changes on the biology of pregnancy. These limitations are not unique to our study and impact virtually all investigations of Hominoid development and disease, particularly the ones of human-specific traits. Endometrial organoids and iPSC-derived ESFs, however, are promising systems in which to study the development of these traits and disease susceptibility that circumvents the limitations of studying human biology (*Abbas et al., 2020*; *Boretto et al., 2017*; *Marinić et al., 2020*; *Rawlings et al., 2021*; *Turco et al., 2017*).

Our gene expression dataset also represents only a snapshot in time of gestation, rather than a comprehensive time course of endometrial gene expression throughout gestation. Interestingly however, the expression changes we identified from these early time points are enriched in disease ontology terms related to adverse pregnancy outcomes that span the length of gestation including infertility, recurrent spontaneous abortion, preeclampsia, and preterm birth. These findings suggest that atypical gene expression patterns and physiological changes at the earliest stages, perhaps even processes occurring in the endometrium before pregnancy (e.g., decidualization of ESFs into DSCs), may predispose to multiple adverse outcomes, including those at the latter stages like preterm birth (birth before 37 weeks). An important focus of future studies should be collecting endometrial samples across species and from multiple stages of pregnancy, particularly close to term, when the mechanisms that maintain gestation cease and those that initiate parturition are likely to be activated.

## Conclusions

We found that hundreds of genes gained or lost endometrial expression in humans, including genes that may contribute to a previously unknown maternal–fetal communication system (*HTR2B*), enhanced

mechanisms for maternal–fetal immunotolerance (*PDCD1LG2* also known as *PD-L2*), and deep placental invasion (*CORIN*). These results demonstrate that gene expression changes at the maternal–fetal interface likely underlie human-specific pregnancy traits and adverse pregnancy outcomes. Our work also illustrates the importance of evolutionary studies for investigating human-specific traits and diseases. This 'evolutionary forward genomics' approach complements traditional forward and reverse genetics in model organisms, which may not be relevant in humans, as well as commonly used methods for characterizing the genetic architecture of disease, such as quantitative trait mapping and GWASs. Specifically, our data demonstrate the importance of evolutionary medicine for a mechanistic understanding of endometrial (dys)function, and suggest that similar studies of other tissue and organ systems will help identify genes underlying normal and pathological anatomy and physiology. We anticipate that our results will further the synthesis of evolution and medicine and may contribute to the development of interventions for adverse pregnancy outcomes such as preterm birth.

# Materials and methods

## Key resources table

| Reagent type (species) or resource | Designation | Source or reference | Identifiers | Additional information |
|---|---|---|---|---|
| Software, algorithm | Kallisto | *Bray et al., 2016* | Version 0.42.4, RRID:SCR_016582 | |
| Software, algorithm | IQ-TREE 2 | *Minh et al., 2020*; *Nguyen et al., 2015* | RRID:SCR_017254 | |
| Software, algorithm | R | | Version 3.6.1 | |
| Software, algorithm | vegan | *Oksanen et al., 2019* | Version 2.5-6, RRID:SCR_011950 | |
| Software, algorithm | Seurat | *Butler et al., 2018* | Version 3.1.1, RRID:SCR_007322 | |
| Software, algorithm | Monocle2 | *Qiu et al., 2017* | Version 2, RRID:SCR_016339 | |
| Software, algorithm | WebGestalt | *Liao et al., 2019* | Version 2019, RRID:SCR_006786 | |
| Antibody | anti-HTR2B (Rabbit polyclonal) | Fisher Scientific | Catalog No. 72-025-6, RRID:AB_2633218 (1:200) | |
| Antibody | IgG (H + L) Cross-Adsorbed Goat anti-Rabbit, Alexa Fluor 594 | Fisher Scientific | Catalog No. A11012, RRID:AB_141359 (1:1000) | |
| Cell line (Human) | T-HESC | ATCC | CRL-4003, RRID:CVCL_C464 | |
| Recombinant DNA reagent | pGL4.29[luc2P/CRE/Hygro] (plasmid) | Promega | E847A | |
| Recombinant DNA reagent | pGL4.44[luc2P/AP1-RE/Hygro] | Promega | E4111 | |
| Recombinant DNA reagent | pGL4.33[luc2P/SRE/Hygro] | Promega | E1340 | |
| Recombinant DNA reagent | pGL4.34[luc2P/SRF-RE/Hygro] | Promega | E1350 | |
| Recombinant DNA reagent | pGL3-Basic[minP] | Promega | E1751; this paper | |

## Endometrial gene expression profiling

Anatomical terms referring to the glandular portion of the female reproductive tract (FRT) specialized for maternal–fetal interactions or shell formation are not standardized. Therefore, we searched the NCBI BioSample, Sequence Read Archive (SRA), and Gene Expression Omnibus (GEO) databases using the search terms 'uterus', 'endometrium', 'decidua', 'oviduct', and 'shell gland' followed by manual curation to identify those datasets that included the region of the FRT specialized for maternal–fetal interaction or shell formation. Datasets that did not indicate whether samples were from pregnant or gravid females were excluded, as were those composed of multiple tissue types. For all RNA-Seq analyses, we used Kallisto (*Bray et al., 2016*) version 0.42.4 to pseudoalign the raw RNA-Seq reads to reference transcriptomes (see *Figure 1—source data 1* for accession numbers and reference genome assemblies) and to generate transcript abundance estimates. We used default

parameters bias correction, and 100 bootstrap replicates. Kallisto outputs consist of transcript abundance estimates in TPM, which were used to determine gene expression. To ensure that human decidua samples were free from trophoblast contamination, we compared the expression of placental enriched genes in RNA-Seq data from human placenta, a human ESF cell line, a human decidual stromal (DSC) cell line, and human first trimester decidua. These results suggest that there is likely no trophoblast contamination of human first trimester decidua samples (*Table 1*), thus inferences of gene expression gains in the human lineage are unlikely to be the result of trophoblast contamination.

Next, we compared two different gene expression metrics to reconstruct the evolutionary history of endometrial gene expression: (1) TPM, a quantitative measure of gene expression that reflects the relative molar ratio of each transcript in the transcriptome; and (2) binary encoding, a discrete categorization of gene expression that classifies genes as expressed (state = 1) or not expressed (state = 0). For binary encoding we transformed transcript abundance estimates into discrete character states, such that genes with TPM ≥2.0 were coded as expressed (state = 1), genes with TPM <2.0 were coded as not expressed (state = 0), and genes without data in specific species coded as missing (state = ?); see *Box 1* for a detailed justification of the TPM ≥2 cutoff. The TPM coded dataset grouped species randomly (*Figure 1—figure supplement 1A*), whereas the binary encoded endometrial gene expression dataset generally grouped species by phylogenetic relatedness (*Figure 1—figure supplement 1B*), suggesting greater signal to noise ratio than raw transcript abundance estimates. Therefore, we used the binary encoded endometrial transcriptome dataset to reconstruct ancestral gene expression states and trace the evolution of endometrial gene expression changes across vertebrate phylogeny (*Figure 1A*). Orthology assessment was inferred using Ensembl Compara.

## Ancestral transcriptome reconstruction

Ancestral states for each gene were inferred with the empirical Bayesian method implemented in IQ-TREE 2 (*Minh et al., 2020*; *Nguyen et al., 2015*) using the species phylogeny shown in *Figure 1A* and the best-fitting model of character evolution determined by ModelFinder (*Kalyaanamoorthy et al., 2017*). The best-fitting model was inferred to be the General Time Reversible model for binary data (GTR2), with character state frequencies optimized by maximum likelihood (FO), and a FreeRate model of among site rate heterogeneity with four categories (R4) (*Soubrier et al., 2012*). We used ancestral transcriptome reconstructions to trace the evolution of gene expression gains (0 → 1) and losses (1 → 0) from the last common ancestor of mammals through to the Hominoid stem-lineage limiting our inferences to reconstructions with BPPs ≥0.80 (*Figure 1A* and *Figure 1—source data 2*). Ancestral reconstructions with BPP ≥0.80 were excluded from over representation analyses.

## Data exploration and MDS

We used classical MDS to explore the structure of extant and ancestral transcriptomes. MDS is a multivariate data analysis method that can be used to visualize the similarity/dissimilarity between samples by plotting data points (in this case transcriptomes) onto two-dimensional plots. MDS returns an optimal solution that represents the data in a two-dimensional space, with the number of dimensions ($k$) specified a priori. Classical MDS preserves the original distance metric, between data points, as well as possible. MDS was performed using the veganR package (*Oksanen et al., 2019*) with four reduced dimensions. Transcriptomes were grouped using $K$-means clustering with $K$ = 2–6, $K$ = 5 optimized the number of distinct clusters and cluster memberships (i.e., correctly grouping species by phylogenetic relationship, parity mode, and placenta type).

## Reanalyses of *Vento-Tormo et al., 2018* endometrial scRNA-Seq data

Maternal–fetus interface 10× Genomics scRNA-Seq data were retrieved from the E-MTAB-6701 entry as a processed data matrix (*Vento-Tormo et al., 2018*). The RNA counts and major cell-type annotations were used as provided by the original publications. Seurat (v3.1.1) (*Butler et al., 2018*), implemented in R (v3.6.0), was used for filtering, normalization, and cell types clustering. The subclusters of cell types were annotated based on the known transcriptional markers from the literature survey. Briefly, we performed the following data processing steps: (1) cells were filtered based on the criteria that individual cells must be expressing at least 1000 and not more than 5000 genes with a count ≥1; (2) cells were filtered out if more than 5 % of counts mapping to mitochondrial genes; (3) data normalization was performed by dividing uniquely mapping read counts (defined by Seurat as unique molecular

identified [UMI]) for each gene by the total number of counts in each cell and multiplying by 10,000. These normalized values were then log-transformed. Cell types were clustered by using the top 2000 variable genes expressed across all samples. Clustering was performed using the 'FindClusters' function with essentially default parameters, except resolution was set to 0.1. The first 20 PCA dimensions were used in the construction of the shared-nearest neighbor (SNN) graph and the generation of two-dimensional embeddings for data visualization using UMAP. Major cell types were assigned based on the original publication samples' annotations, and cell subtypes within major cell types were annotated using the subcluster markers obtained from the above parameters. We then chose the decidual and PV cells to perform the single-cell trajectory, pseudotime analysis, and cell ordering along an artificial temporal continuum analysis using Monocle2 (*Qiu et al., 2017*). The top 500 differentially expressed genes were used to distinguish between the subclusters of decidua and PV cell populations on pseudotime trajectory. The transcriptome from every single cell represents a pseudo-time point along an artificial time vector that denotes decidual and PV lineages' progression, respectively. To compare the differentially expressed genes between *HTR2B*-positive and *HTR2B*-negative cells, we first divided the decidual and PV datasets into those groups of cells that either express *HTR2B* with a count ≥1 and those with zero counts. We then performed differentially expressed genes analysis between the mentioned two groups of cells using the bimodal test for significance.

To calculate the enrichment score of human-gain genes in each cell type, we first transformed the data into a pseudobulk expression matrix by averaging all genes' expression in each cell type. We then calculated the fraction of human-gained genes expressed (Observed) and the proportion of the rest of the genes expressed in each cell type (Expected). The enrichment ratio shown on the plot is the ratio of Observed and Expected values for each cell type. The p value was calculated using a two-way Fisher exact test followed by Bonferroni correction.

## Reanalyses of *HTR2B*, *PDCD1LG2*, and *CORIN* expression across multiple endometrial scRNA-Seq datasets

Transcriptomic dynamics of human endometrium in vivo. Data mined from publicly available database reproductivecellatlas.org (*Garcia-Alonso et al., 2021*). Data show a cellular map of the human endometrium from combinatorial transcriptomics (scRNA-Seq and single-nuclei RNA sequencing [snRNA-Seq]) alongside spatial transcriptomics methods (10× Genomics Visium slides and high-resolution microscopy) representing 98,568 cells from fifteen individuals grouped into five main cellular types. No reuse allowed without permission.

Single-cell analysis of peri-implantation endometrium. Six LH-timed endometrial biopsies were processed for Droplet generation and single-cell sequencing (Drop-Seq) as described in *Lucas et al., 2020*. Anonymized endometrial biopsies were obtained from women aged between 31 and 42 years with regular cycles, body mass index between 23 and 32 kg/m$^2$, and the absence of uterine pathology on transvaginal ultrasound examination. *t*-Distributed stochastic neighbour embedding (*t*-SNE) analysis assigned 2831 cells to four clusters, designated based on canonical marker genes as ECs (*n* = 141), epithelial cells (EpC; *n* = 395), immune cells (IC; *n* = 352), and ESFs (*n* = 1943). Data are available in the GEO repository GSE127918.

### Deconvolution of in vitro cell types from endometrium

Three independent midluteal biopsies were used to isolate, culture, and sequence different endometrial cell types. Following enzymatic digestion, EpC were separated from the stromal cell fraction as described (*Barros et al., 2016*). PVCs and ESFs were then subjected to magnetic activated cell sorting (MACS) using W5C5 antibody (*Masuda et al., 2012*). PVC and ESFs were maintained in standard cultures as well as subjected to colony-forming unit assays. Total RNA was then extracted from the resulted clones, designated endometrial mesenchymal stem cells (eMSCs) and transit amplifying cells (TA), respectively. The standard PVC and ESF cultures were propagated until 90 % confluence and then subjected to total RNA extractions. Primary EpC were subjected to gland organoid formation (*Turco et al., 2017*). uNK cells were also isolated although not cultured. After overnight incubation, the supernatant of the stromal cell fraction was subjected to MACS to isolate uNK cells using a PE-conjugated antihuman CD56 monoclonal antibody. Libraries were prepared using TruSeq RNA Library preparation kit V2 and sequenced on HiSeq 4000 with 75 bp paired-end reads. Data are presented as TPM (*Diniz-da-Costa et al., 2021* [unpublished thesis]).

## Endometrium through the menstrual cycle

Data mined from the publicly available database GDS2052 (*Talbi et al., 2006*). Data are presented as average gene counts.

## Single-cell analysis of the decidual pathway in vitro

Primary ESFs were decidualized with a progestin (medroxyprogesterone acetate, MPA) and a cyclic adenosine monophosphate analog (8-bromo-cAMP, cAMP) for 8 days. Cells were recovered every 48 hr and subjected to single-cell analysis using nanoliter droplet barcoding and high-throughput RNA sequencing. Approximately 800 cells were sequenced per time point, yielding on average 1282 genes per cell. After computational quality control 4580 cells were assigned to 7 transcriptional cell states (6 presented) using SNN and *t*-SNE methods and presented as transcriptional states. Data are available in the GEO repository GSE127918. Data are presented as TPM (*Lucas et al., 2020*).

## Endometrial stromal lineage cell nomenclature

*Vento-Tormo et al., 2018* dataset consists of transcriptomes for ~70,000 individual cells of many different cell types, including: three populations of tissue resident decidual natural killer cells (dNK1, dNK2, and dNK3), a population of proliferating natural killer cells (dNKp), type two and/or type three ILCs (ILC2/ILC3), three populations of decidual macrophages (dM1, dM2, and dM3), two populations of DCs (DC1 and DC2), granulocytes (Gran), T cells (TCells), maternal and lymphatic endothelial cells (Endo), two populations of epithelial glandular cells (Epi1 and Epi2), two populations of PVCs (PV1 and PV2), two ESF populations (ESF1 and ESF2), and DSCs, placental fibroblasts (fFB1), extravillous- (EVT), syncytio- (SCT), and villus- (VCT) cytotrophoblasts (*Figure 3* and *Figure 3—figure supplements 1 and 2*).

We note that Vento-Tormo et al. identified five populations of cells in the endometrial stromal lineage, including two perivascular populations (likely reflecting the mesenchymal stem cell-like progenitor of ESFs and DSCs) and three cell types they call 'decidual stromal cells' and label 'S1–3'. However, based on the gene expression patterns of 'dS1–3' (shown in Vento-Tormo et al. *Figure 3a*), only 'dS3' are decidualized, as indicated by expression of classical markers of decidualization such and *PRL* (*Tabanelli et al., 1992*) and *IGFBP1/2/6* (*Tabanelli et al., 1992*; *Kim et al., 1999*). In stark

**Table 1.** Expression of placental enriched genes in RNA-Seq data from human placenta, endometrial stromal fibroblasts (ESFs), decidual stromal cells (DSCs), and human first trimester decidua. Expression levels are shown as transcripts per million (TPM) values, the Tissue Specificity (TS) score is calculated as the fold enrichment of each gene relative to the tissue with the second highest expression of that gene. Placental data are from https://www.proteinatlas.org/humanproteome/tissue/placenta.

| Gene | Description | TS | Placenta | ESF | DSC | Decidua |
|------|-------------|-----|----------|-----|-----|---------|
| CSH1 | Chorionic somatomammotropin hormone 1 | 211 | 13,487 | 0 | 0 | 0 |
| CSH2 | Chorionic somatomammotropin hormone 2 | 193 | 3932 | 0 | 0 | 6.67 |
| CSHL1 | Chorionic somatomammotropin hormone like 1 | 352 | 217 | 0 | 0 | 0 |
| GH2 | Growth hormone 2 | 199 | 189 | 0 | 0 | 0.6 |
| HBG1 | Hemoglobin subunit gamma 1 | 147 | 18,254 | 0 | 0 | 0 |
| ISM2 | Isthmin 2 | 112 | 274 | 0 | 0.2 | 1.5 |
| PSG1 | Pregnancy-specific beta-1-glycoprotein 1 | 377 | 363 | 0.64 | 20.9 | 0 |
| PSG2 | Pregnancy-specific beta-1-glycoprotein 2 | 222 | 343 | 0.15 | 7.9 | 0.5 |
| PSG3 | Pregnancy-specific beta-1-glycoprotein 3 | 162 | 189 | 0 | 0.6 | 0.7 |
| PSG5 | Pregnancy-specific beta-1-glycoprotein 5 | 108 | 158 | 2.3 | 14.3 | 0 |
| PSG9 | Pregnancy-specific beta-1-glycoprotein 9 | 118 | 147 | 47.2 | 36.1 | 0.0 |
| XAGE3 | X antigen family member 3 | 167 | 575 | 0 | 0 | 0.05 |

contrast, 'dS1' do not express decidualization markers but highly express markers of ESFs such as *TAGLN* and *ID2*, as well as markers of proliferating ESFs including *ACTA2* (*Kim et al., 1999*). 'dS2' also express ESFs markers (*TAGLN*, *ID2*, *ACTA2*), but additionally *LEFTY2* and *IGFBP1/2/6*, consistent with ESFs that have initiated the process of decidualization. These data indicate that the 'dS1' and 'dS2' populations are both ESFs, but 'dS2' are ESFs that have initiated decidualization (because they express *IGFBPs* but not *PRL*), and that 'dS3' are DSCs. Vento-Tormo et al. show that the differences in gene expression between 'dS1–3' are related to their topography in the endometrium, but degree of decidualization ('dS1'/ESF1 < 'dS2'/ESF2 < 'dS3'/DSC) is also linked to differential gene expression.

Consistent with this, other scRNA-Seq studies have identified two ESF populations and one DSC population in the first trimester decidua, and used pseudotime analyses to show that they represent different states of differentiation from ESFs to mature DSCs (*Suryawanshi et al., 2018*). Therefore, we prefer to use the perivascular/ESF/DSC nomenclature because it more accurately reflects the biology and gene expression profile of these cell types than the 'dS1–3' naming convention. We also note that while it is generally thought that ESFs are absent from the pregnant uterus, ESFs retain a presence in the endometrium from the first trimester until term (*Richards et al., 1995*; *Suryawanshi et al., 2018*; *Muñoz-Fernández et al., 2019*; *Sakabe et al., 2020*).

## Overrepresentation analyses

We used WebGestalt v. 2019 (*Liao et al., 2019*) to identify enriched ontology terms using overrepresentation analysis (ORA). We used ORA to identify enriched terms for three pathway databases (KEGG, Reactome, and Wikipathway), three disease databases (Disgenet, OMIM, and GLAD4U), and a custom database of genes implicated in preterm birth by GWAS. The preterm birth gene set was assembled from the NHGRI-EBI Catalog of published GWASs (GWAS Catalog), including genes implicated in GWAS with either the ontology terms 'Preterm Birth' (EFO_0003917) or 'Spontaneous Preterm Birth' (EFO_0006917), as well as two recent preterm birth and birth weight GWASs (*Warrington et al., 2019*; *Sakabe et al., 2020*) using a genome-wide significant p value of $9 \times 10^{-6}$. The custom gmt file used to test for enrichment of preterm birth associated genes is included as a supplementary data file to (*Figure 2*, *Figure 2—source data 1*).

## Functional genomic analyses of the *HTR2B*, *PDCD1LG2*, and *CORIN* loci

We used previously published ChIP-Seq data generated from human DSCs that were downloaded from NCBI SRA and processed remotely using Galaxy (*Afgan et al., 2016*). ChIP-Seq reads were mapped to the human genome (GRCh37/hg19) using HISAT2 (*Kim et al., 2019*; *Kim et al., 2015*; *Pertea et al., 2016*) with default parameters and peaks called with MACS2 (*Feng et al., 2012*; *Zhang et al., 2008*) with default parameters. Samples included PLZF (GSE75115), H3K4me3 (GSE61793), H3K27ac (GSE61793), H3K4me1 (GSE57007), PGR (GSE69539), the PGR-A and -B isoforms (GSE62475), NR2F2 (GSE52008), FOSL2 (GSE69539), FOXO1 (GSE69542), PolII (GSE69542), GATA2 (GSE108408), SRC-2/NCOA2 (GSE123246), AHR (GSE118413), ATAC-Seq (GSE104720), and DNase1-Seq (GSE61793). FAIRE-Seq peaks were downloaded from the UCSC genome browser and not recalled.

We also used previously published RNA-Seq and microarray gene expression data generated from human ESFs and DSCs that were downloaded from NCBI SRA and processed remotely using Galaxy platform (https://usegalaxy.org/; Version 20.01) for RNA-Seq data and GEO2R for microarray data. RNA-Seq datasets were transferred from SRA to Galaxy using the Download and Extract Reads in FASTA/Q format from NCBI SRA tool (version 2.10.4+ galaxy1). We used HISAT2 (version 2.1.0+ galaxy5) to align reads to the Human hg38 reference genome using single- or paired-end options depending on the dataset and unstranded reads, and report alignments tailored for transcript assemblers including StringTie. Transcripts were assembled and quantified using StringTie (v1.3.6)(*Pertea et al., 2016*; *Pertea et al., 2015*), with reference file to guide assembly and the 'reference transcripts only' option, and output count files for differential expression with DESeq2/edgeR/limma-voom. Differentially expressed genes were identified using DESeq2 (*Love et al., 2014*) (version 2.11.40.6+ galaxy1). The reference file for StringTie guided assembly was wgEncodeGencodeBasicV33. GEO2R performs comparisons on original submitter-supplied processed data tables using the GEOquery and limma R packages from the Bioconductor project (https://bioconductor.org/). These datasets included gene expression profiles of primary human ESFs treated for 48 hr with control nontargeting, PGR-targeting (GSE94036), FOXO1-targeting (GSE94036), or NR2F2 (COUP-TFII)-targeting (GSE47052)

siRNA prior to decidualization stimulus for 72 hr; transfection with GATA2-targeting siRNA was followed immediately by decidualization stimulus (GSE108407). Probes were 206638_at (*HTR2B*), 220049_s_at (*PDCD1LG2*), and 220356_at (*CORIN*) for GSE4888 (endometrial gene expression throughout menstrual cycle) and for GSE5999 (gene expression in basal plate throughout gestation). Multispecies RNA-Seq analysis of ESFs and DSCs is from GSE67659.

To assess chromatin looping, we utilized a previously published H3K27ac HiChIP dataset from a normal hTERT-immortalized endometrial cell line (E6E7hTERT) and three endometrial cancer cell lines (ARK1, Ishikawa, and JHUEM-14) (*O'Mara et al., 2019*).

## Immunofluorescent staining for endometrial HTR2B

Endometrial biopsies were fixed overnight in 10 % neutral buffered formalin at 4 °C and wax embedded in Surgipath Formula 'R' paraffin using the Shandon Excelsior ES Tissue processor (Thermo Fisher). Tissues were sliced into 3 µm sections on a microtome and adhered to coverslips by overnight incubation at 60 °C. Deparaffinization and rehydration were performed through xylene, 100 % isopropanol, 70 % isopropanol, and distilled water incubations. Following antigen retrieval, slides were washed, blocked, and incubated in primary HTR2B antibody (1:200; Fisher Scientific) overnight at 4 °C. After washing three times, slides were incubated with Alexa Fluor 594 (1:1000; Fisher Scientific) for 2 hr, washed and mounted in ProLong Gold → Antifade Reagent with DAPI (Cell Signaling Technology). Slides were visualized using the EVOS Auto system, with imaging parameters maintained throughout image acquisition.

## Cell culture and serotonin (5-HT) treatment

Human hTERT-immortalized endometrial stromal fibroblasts (T-HESC; CRL-4003, ATCC) were grown in maintenance medium, consisting of Phenol Red-free DMEM (31053-028, Thermo Fisher Scientific), supplemented with 10 % charcoal-stripped fetal bovine serum (12676029, Thermo Fisher Scientific), 1 % L-glutamine (25030-081, Thermo Fisher Scientific), 1 % sodium pyruvate (11360070, Thermo Fisher Scientific), and 1× insulin–transferrin–selenium (ITS; 41400045, Thermo Fisher Scientific).

A total of $10^4$ ESFs were plated per well of a 96-well plate, 18 hr later cells were transfected in Opti-MEM (31985070, Thermo Fisher Scientific) with 100 ng of luciferase reporter plasmid, 10 ng Renilla control plasmid, 0.25 µl of Lipofectamine LTX (15338100, Thermo Fisher Scientific) and 0.1 µl Plus Reagent as per the manufacturer's protocol per well; final volume per well was 100 µl. Luciferase reporter plasmids were synthesized (GenScript) by cloning the response elements from the pGL4.29[luc2P/CRE/Hygro], pGL4.44[luc2P/AP1-RE/Hygro], pGL4.33[luc2P/SRE/Hygro], and pGL4.34[luc2P/SRF-RE/Hygro] plasmids into pGL3-Basic[minP] luciferase reporter. Unlike the pGL4 series vectors (Promega) that are hormone responsive, pGL3-Basic[minP] luciferase reporter includes a minimal promoter but is not hormone responsive. Final pathway reporter plasmids are: CRE_pGL3-Basic[minP] (cAMP/PKA), AP1_pGL3-Basic[minP] (AP1), SRE_pGL3-Basic[minP] (MAPK/ERK), and SRF_RE_pGL3-Basic[minP] (serum response factor).

ESFs were incubated in the transfection mixture for 6 hr. Then, ESFs were washed with warm PBS and incubated in the maintenance medium overnight. The next day, the medium in half of the wells was exchanged for the differentiation medium consisting of DMEM with Phenol Red and GlutaMAX (10566-024, Thermo Fisher Scientific), supplemented with 2 % fetal bovine serum (26140-079, Thermo Fisher Scientific), 1 % sodium pyruvate (11360070, Thermo Fisher Scientific), 1 µM medroxyprogesterone 17-acetate (MPA; M1629, Sigma Aldrich), and 0.5 mM 8-Bromoadenosine 3′,5′-cyclic monophosphate (8-Br-cAMP; B5386, Sigma Aldrich). After 48 hr, serotonin (5-HT; H9523, Sigma Aldrich) was added to the wells with both maintenance and differentiation medium (for each in triplicates) in the following concentrations: 50 µM, 200 µM, and 1 mM; vehicle control (0 µM) was water. After 6 hr of incubation, we used a Dual Luciferase Reporter Assay (Promega) to quantify luciferase and Renilla luminescence following the manufacturer's Dual Luciferase Reporter Assay protocol.

## Cell lines

Human hTERT-immortalized endometrial stromal fibroblasts were purchased from ATCC (T-HESC; CRL-4003, ATCC). Their identity has been authenticated by ATCC, and was determined to be mycoplasma free.

## Acknowledgements

This study was supported by a grant from the March of Dimes (March of Dimes Prematurity Research Center) and a Burroughs Welcome Fund Preterm Birth Initiative grant (1013760) to principal investigator VJL. The Tommy's Charity funds the Tommy's National Centre for Miscarriage Research to JJB. JJB holds a Wellcome Trust Investigator Award (212233/Z/18/Z). MS was supported by a presidential postdoctoral fellowship from Cornell University. The funders had no role in study design, data collection and analysis, decision to publish, or preparation of the manuscript.

## Additional information

### Funding

| Funder | Grant reference number | Author |
| --- | --- | --- |
| March of Dimes Foundation | Prematurity Research Center | Vincent J Lynch |
| Burroughs Wellcome Fund | 1013760 | Vincent J Lynch |
| Wellcome Trust | 212233/Z/18/Z | Jan Joris Brosens |

The funders had no role in study design, data collection and interpretation, or the decision to submit the work for publication.

### Author contributions

Katelyn Mika, Data curation, Formal analysis, Writing – original draft; Mirna Marinić, Conceptualization, Data curation, Formal analysis, Methodology, Writing – original draft; Manvendra Singh, Conceptualization, Data curation, Formal analysis, Investigation, Writing – original draft; Joanne Muter, Data curation, Formal analysis, Investigation, Visualization, Writing – review and editing; Jan Joris Brosens, Conceptualization, Formal analysis, Supervision, Visualization, Writing – review and editing; Vincent J Lynch, Conceptualization, Data curation, Formal analysis, Funding acquisition, Investigation, Supervision, Visualization, Writing – original draft, Writing – review and editing

### Author ORCIDs

Katelyn Mika http://orcid.org/0000-0002-2170-9364
Mirna Marinić http://orcid.org/0000-0002-7037-8389
Manvendra Singh http://orcid.org/0000-0002-8626-5418
Jan Joris Brosens http://orcid.org/0000-0003-0116-9329
Vincent J Lynch http://orcid.org/0000-0001-5311-3824

### Decision letter and Author response

Decision letter https://doi.org/10.7554/eLife.69584.sa1
Author response https://doi.org/10.7554/eLife.69584.sa2

## Additional files

### Supplementary files
• Transparent reporting form
• Source data 1. Enrichment result table for *Box 2—figure 1*.

### Data availability
All gene expression data analysed during this study are publicly available, accession numbers of given in Figure 1 - source data 1.

The following previously published datasets were used:

| Author(s) | Year | Dataset title | Dataset URL | Database and Identifier |
|---|---|---|---|---|
| Roost MS, van Iperen L, Ariyurek Y, Buermans HP, Arindrarto W, Devalla HD, Passier R, Mummery CL, Carlotti F, de Koning EP, van Zwet EW, Goeman JJ, SousaLopes SM | 2015 | KeyGenes, a Tool to Probe Tissue Differentiation Using a Human Fetal Transcriptional Atlas | https://www.ncbi.nlm.nih.gov/geo/query/acc.cgi?acc=GSE66302 | NCBI Gene Expression Omnibus, GSE19373 |
| Mika KM, Lynch VJ | 2021 | Evolutionary transcriptomics implicates HAND2 in the origins of implantation and regulation of gestation length | https://www.ncbi.nlm.nih.gov/geo/query/acc.cgi?acc=GSE155170 | NCBI Gene Expression Omnibus, GSE155170 |
| Yang Z, Liu J | 2012 | oint analysis of microRNome and 3'-UTRome in the endometrium of rhesus monkey | https://www.ncbi.nlm.nih.gov/geo/query/acc.cgi?acc=GSE31041 | NCBI Gene Expression Omnibus, GSE31041 |
| Liu J | 2015 | Identification of gene expression changes in rabbit uterus during embryo implantation | https://www-ncbi-nlm-nih-gov.ezproxy.u-pec.fr/geo/query/acc.cgi?acc=GSE76115 | NCBI Gene Expression Omnibus, GSE76115 |
| Woods L, Perez-Garcia V, Kieckbusch J, Wang X, DeMayo F, Colucci F, Hembrger M | 2017 | Decidualisation and placentation defects are a major cause of age-related reproductive decline | https://www-ncbi-nlm-nih-gov.ezproxy.u-pec.fr/geo/query/acc.cgi?acc=GSE98901 | NCBI Gene Expression Omnibus, GSE98901 |

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
