## [Decision Letter]

**Acceptance summary:**

The study by Mika and colleagues uses a comparative transcriptomics approach to identify changes in the expression of genes that specifically occurred during the evolution of the human endometrium. The authors find that hundreds of genes gained or lost endometrial expression in the human lineage and that several of these genes are potentially implicated in the pathophysiology of human pregnancy. The study contributes to ongoing interest in the effect of human evolution on the pathophysiology of human pregnancy, and has the potential to serve as a model of how to study the evolution of pregnancy-associated genomic changes in particular species and tissues.

**Decision letter after peer review:**

Thank you for submitting your article "Evolutionary transcriptomics implicates new genes and pathways in human pregnancy and adverse pregnancy outcomes" for consideration by *eLife*. Your article has been reviewed by 3 peer reviewers, and the evaluation has been overseen by a Reviewing Editor and George Perry as the Senior Editor. The following individuals involved in review of your submission have agreed to reveal their identity: Adam Stevens (Reviewer #1); Derek E Wildman (Reviewer #2); Sam Mesiano (Reviewer #3).

Essential revisions:

1) Please ensure that your statements accurately reflect your findings. There are several comments by the reviewers on this issue that you should address.

2) Please address reviewer #3's comment on the uncertain quality of the source transcriptome data.

3) Please address reviewer #1's comment that the comparison of a receptive and non-receptive endometrium has not been fully accounted for.

4) Please compare the Vento-Tormo scRNAseq data set (Nature 2018, 563:347-353) with the term maternal-fetal interface scRNAseq data set produced by the Gomez-Lopez group (Pique-Regi et al. *eLife* 2019;8:e52004), as suggested by reviewer #3.

*Reviewer #1 (Recommendations for the authors):*

Specifically, can the authors comment on the impact of endometrial receptivity and diapause on their algorithm to identify gene expression under the impact of evolution?

I accept that the quality and nature of the downstream analysis using other omic data sets is in itself good supporting evidence that the alorithm used to determine evolutionary impact must be working but there is room for additional comment and streamlined usage.

It would seem to me that establishing thoroughly the utility of this algorithm is key to entire package of work.

*Reviewer #3 (Recommendations for the authors):*

1) It is not clear why/how the 3 genes of interest were selected. In UMPAs shown in Figure 7 and 8 very few DSCs are positive for CORIN and PDCD1LG2. The same outcomes are obtain using the online dashboards for the Vento-Tormo (https://maternal-fetal-interface.cellgeni.sanger.ac.uk) and Pique-Regi (http://placenta.grid.wayne.edu) data sets. How do these data justify these as endometrial cell genes of interest? In contrast it is clear that HTR2B is expressed in DSCs. However, the word cloud in Figure 2 shows only minor emphasis for serotonin receptor signaling.

2) The transcriptome data from non pregnancy ESFs ± decidualization in vitro for 72h may not be a valid comparison with the pregnancy endometrium because the cells are not subjected to paracrine signaling from adjacent fetal cells.

3) How was the promoter element for PGR-A identified? Is it distinct from the PGR-B promoter element? Do the PGR isoforms even have specific promoter motif targets.

4) Experiments described for Figure 6F lacks positive controls for the luciferase reporter vectors. There is no verification that the cells were transfected with functional reporters. The negative data are therefore meaningless without positive controls for each reporter.

5) The study would be strengthened by more comprehensive validation of expression (e.g., IHC) of the 3 genes of interest in human decidua basalis and parietals.

6) It is stated that "a role for decidual PDCD1LG2 in pregnancy is strongly suggested by its association with variants linked to gestational length and number of lifetime pregnancies (parity)". The data related to this statement should be cited.

7) How was the data in Figure 1 (amniote phylogeny and MDS analyses) calculated. This is an interesting figure but as presented and described the point is lost on this reviewer.

8) The UMAP in Figure 3 is markedly different to that shown in the Vento-Tormo paper and shows 5 DSC phenotypes vs 3. Please explain the difference? Also, evCTBs are included in the cells with significantly enriched recruited genes. Does this reflect contamination by these cells in analyses used to identify recruited genes in endometrial cells?

9) Figure 4A-C are meaningless to this reviewer. Large amount of data but little increment in knowledge.

10) Please cite the source data for Figure 5A and B in the Figure legend. What is meant by minus 0 in the lower panel of Figure 5F? Figure 5F could be omitted.

---

## [Author Response]

Reviewer #1 (Recommendations for the authors):Specifically, can the authors comment on the impact of endometrial receptivity and diapause on their algorithm to identify gene expression under the impact of evolution?

Although we have only included samples from pregnant endometria, and therefore there is no effects of differential receptivity or diapause on our analyses, it is possible to extend our evolutionary analyses to these states. The comment about diapause is very interesting, and is one we intend to study using armadillo and wallaby.

I accept that the quality and nature of the downstream analysis using other omic data sets is in itself good supporting evidence that the algorithm used to determine evolutionary impact must be working but there is room for additional comment and streamlined usage.It would seem to me that establishing thoroughly the utility of this algorithm is key to entire package of work.

We believe the “algorithm used to determine evolutionary impact” the reviewer is referring to is ancestral sequence reconstruction. While methods for ancestral sequence reconstruction are very well established within comparative and evolutionary biology, we incorrectly took for granted that readers from other fields would be familiar with these methods. We now include two citations in our Results section where we describe the ancestral sequence reconstruction method, one to the (likely) original paper describing the method (Pauling and Zukerkandl 1963) and an excellent review of the methods (Joy et al., 2016).

Reviewer #3 (Recommendations for the authors):1) It is not clear why/how the 3 genes of interest were selected. In UMPAs shown in Figure 7 and 8 very few DSCs are positive for CORIN and PDCD1LG2. The same outcomes are obtain using the online dashboards for the Vento-Tormo (https://maternal-fetal-interface.cellgeni.sanger.ac.uk) and Pique-Regi (http://placenta.grid.wayne.edu) data sets. How do these data justify these as endometrial cell genes of interest? In contrast it is clear that HTR2B is expressed in DSCs. However, the word cloud in Figure 2 shows only minor emphasis for serotonin receptor signaling.

The primary goal of this study was to determine if an evolutionary approach to reconstructing the origin and divergence of gene expression patterns in tissue and organ systems could provide unique and useful information about how those systems function, and if so, how dysfunction in those gene expression patterns might contribute to organ system dysfunction. Therefore, we selected three genes that had not previously been implicated in the function of ESFs/DSCs and pregnancy based on pathway enrichments.

It is true that only a subset of ESFs and DSCs express CORIN (related to the enriched term “pre-eclampsia”) and PDCD1LG2 (related to multiple enriched immune system terms). These data indicate that sub-population of endometrial stromal lineage cells express these genes and suggest that there is functional variation within ESFs/DSCs. The concern that these genes are only expressed in a subset (although they are highly expressed in that subset) is one of the reasons we included functional genomics data of these loci, and include a detailed characterization of endometrial stromal lineage development shown in Figures 3 and 4 (see also our response to reviewer critique #9 below). We also note that the proportion of ESFs/DSCs that express PDCD1LG2 is similar to the proportion of macrophage and dendritic cells that express PDCD1LG2, suggesting that sub-populations of several cell-types express this gene rather than all cells within a cell-type.

While the serotonin signaling pathway appears to have only a minor emphasis in the word cloud in Figure 2, that is an effect of the way word clouds represent term size – recruited genes are significantly enriched in the serotonin receptor signaling pathway (P = 0.00043) and the enrichment is quite large (13.247x). HTR2B was selected because it is a gene in this pathway and a role for serotonin signaling in the endometrium has not been reported (other than the early references we cite).

2) The transcriptome data from non pregnancy ESFs ± decidualization in vitro for 72h may not be a valid comparison with the pregnancy endometrium because the cells are not subjected to paracrine signaling from adjacent fetal cells.

We agree with the reviewer that ESF/DSC monoculture does not even come close to recapitulating the complexity of the endometrium during pregnancy. However, the goal of the in vitro experiments that decidualize ESFs into DSCs is to identify which genes are progesterone responsive rather than identify how the expression of genes in DSCs changes in response to the complex paracrine signaling in the endometrium during pregnancy.

3) How was the promoter element for PGR-A identified? Is it distinct from the PGR-B promoter element? Do the PGR isoforms even have specific promoter motif targets.

After careful consideration we believe this question is related to PGR-A and PGR-B binding sites in the genome rather than promoter elements for PGR gene. PGR-A and -B only differ in the presence of a 165 amino acid activation function domain (AF3) in PGR-B that is absent from PGR-A; these isoforms result from differential promoter and transcription start site usage of the PGR gene. Although it is possible to identify PGR-B in experimental assays that use an antibody to the AF3 domain, it is not possible to identify PGR-A because there is no unique epitope in PGR-A. This structural difference also makes it impossible to differentiate PGR-A and -B binding sites in ChIP-seq experiments that use an antibody which recognizes the common part of the protein isoforms. Thus, ChIP-seq experiments using endogenous PGR isoforms, such as the Mazur et al. (2015) ChIP-seq dataset, cannot differentiate PGR-A and -B binding sites. In contrast, Kaya et al. (2015) designed an experiment to differentiate PGR-A and PGR-B binding sites. Specifically, Kaya et al. (2015) expressed PGR-A and PGR-B individually after silencing endogenous PGR in human DSCs, which allowed for differentiation between binding sites of these isoforms by ChIP-seq; PGR-A and -B binding motifs are identical in Kaya et al. (2015).

4) Experiments described for Figure 6F lacks positive controls for the luciferase reporter vectors. There is no verification that the cells were transfected with functional reporters. The negative data are therefore meaningless without positive controls for each reporter.

We apologize that this figure was unclear. The dual luciferase reporter assays shown in Figure 6F and Figure 6—figure supplement 2 do have positive controls, specifically cells transfected with the luciferase (Luc) pathway reporter (and Renilla (Ren)) but without serotonin treatment. In these experiments, which are standardized to non-transfected cells, Luc/Ren values are greater than background. For example, luminescence is ~40x greater than background in Figure 6F, which shows cells transfected with the Luc pathway reporter Ren but without serotonin treatment. These data indicate that cells were transfected with functional reporters and that negative data are reliable. Our main finding from these experiments is that decidual stromal cells are responsive to serotonin treatment, which is shown in Figure 6F and not related to negative findings shown in Figure 6—figure supplement 2. Thus, our conclusion from these experiments, that DSCs are sensitive to serotonin, hold even if it true that we lacked verification that the cells were transfected with functional reporters.

5) The study would be strengthened by more comprehensive validation of expression (e.g., IHC) of the 3 genes of interest in human decidua basalis and parietals.

We agree and part of our delay in submitting this revision was because we were attempting to collect these samples. Unfortunately, we were unable to do so for various reasons. Instead we have explored HTR2B expression in LH-timed (day 6, 7, and 9) endometrial biopsies because these tissue samples are available and we focus most of our validation on this gene. The immunofluorescence images show that HTR2B is diffusely localized in the cytoplasm of endometrial cells as well as glandular cells and is also intensely localized on the luminal surface of the gland cells. These results show that HTR2B protein is present in the endometrium around the window of implantation, but further studies are required to answer the important question of where HTR2B is localized in human decidua basalis and parietals.

6) It is stated that "a role for decidual PDCD1LG2 in pregnancy is strongly suggested by its association with variants linked to gestational length and number of lifetime pregnancies (parity)". The data related to this statement should be cited.

References added.

7) How was the data in Figure 1 (amniote phylogeny and MDS analyses) calculated. This is an interesting figure but as presented and described the point is lost on this reviewer.

Figure 1 shows the rate of gene expression evolution (as gain and losses per gene) in panel A and overall similarity of gene expression in different species in the MDS plot in panel B. The phylogeny in Figure 1A is based on other data and not inferred from endometrial gene expression data, however, branch lengths (and ancestral states) are inferred from the endometrial gene expression data. The take home message from this figure is that internal branch lengths are short while terminal lengths are long indicating that there is a lot of species-specific gains and losses of gene expression in the pregnant endometrium.

The MDS plot in Figure 1B was generated using the magrittr, dplyr, and ggpubr R libraries, and clusters identified using K-means clustering. While conceptually similar to a principal component analysis (PCA), (non-metric) multidimensional scaling (MDS) does not require preprocessing, unlike linear methods such as PCA (PMID: 15509613). The take home message from this plot is that species cluster by their overall gene expression profiles. In this case, the MDS plot shows that primates form a distinct cluster from the other Eutherians and even other eutherians with invasive, hemochorial placentas. Thus, primate gene expression profiles are distinct from other species highlighting that endometrial gene expression in these species has diverged significantly from other species. The matrix used to generate the plot is provided as Figure 1 – Source data 3. Binary encoded matrix of gene expression from extant and ancestral reconstructions used to generate figure 1B.

8) The UMAP in Figure 3 is markedly different to that shown in the Vento-Tormo paper and shows 5 DSC phenotypes vs 3. Please explain the difference?

The difference between the UMAP shown in Vento-Tormo et al. (Nature 2018, 563:347-353) and the UMAPs we are due to use of different versions of Seurat, in our case Seurat (v3.1.1) while the UMAP shown in Vento-Tormo et al. (2018) Seurat (v2.3.3). We note that neither of these UMAPs are “right” or “correct” they simply reflect differences in parameters between the UMAP algorithm.

While we appreciate the concerns of the reviewer with regard to our identification of five DSC populations compared to three in Vento-Tormo et al. (2018), the naming used by Vento-Tormo et al. (2018) is based on the location of these three cell populations in the decidua and does not reflect their cell-type identity. Vento-Tormo et al. also identify five populations of cells in the endometrial stromal lineage, but assign them different names than we do. Specifically, they identify two perivascular populations (likely reflecting the mesenchymal stem cell like progenitor of endometrial stromal fibroblasts and decidual stromal cells) and three cell-types they call “decidual stromal cells” and label “dS1-3”. However, based on the gene expression patterns of “dS1-3” (shown in Vento-Tormo et al. Figure 3a) only “dS3” are decidualized, as indicated by expression of classical markers of decidualization such and *PRL* (PMID: 1534990) and *IGFBP1/2/6* (PMID: 1534990, PMID: 10690800). In stark contrast, “dS1” do not express markers of decidualization but highly express markers of endometrial stromal fibroblasts (ESFs) such as *TAGLN* and *ID2,* as well as markers of proliferating ESFs including *ACTA2* (PMID: 10690800). Like the “dS1” population, “dS2” express markers of proliferating ESFs (*TAGLN*, *ID2*, *ACTA2*) but also express markers of decidualized cells such as *LEFTY2* and *IGFBP1/2/6*, consistent with ESFs that have initiated the process of decidualization. These data indicate that the “dS1” and “dS2” populations are ESFs, that “dS2” are ESFs which have initiated decidualization (because they express *IGFBPs* but not *PRL*), and that “dS3” are DSCs.

Vento-Tormo et al. show that the differences in gene expression between “dS1-3” are related to their topography in the endometrium, but these gene expression differences *also* reflect differences in their degree of decidualization from ESF1 (“dS1”) -> ESF2 (“dS2”) -> DSC (“dS3”). Consistent with this, other scRNA-Seq studies have also identified two ESF populations and one DSC population in first trimester decidua, and used pseudotime analyses to show that they represent different states of differentiation from ESFs to mature DSCs (PMCID: PMC6209386). Thus, like us, Vento-Tormo et al. (2018) identify five DSC populations, but name these populations slight differently than our naming convention. The function and cell-type identity of the five populations we identify are the same, but we prefer to use the ESF1/ESF2/DSC nomenclature because it more accurately reflects the biology and gene expression profile of these cell-types compared to the “dS1-3” naming convention.

We appreciate that changing the terminology for these cell-types can be confusing, which was part of the reason we included Figure 4A-C to ensure that readers understood which endometrial stromal lineage cells we discuss throughout the manuscript. To ensure that this is clear, we now include a section in the methods that describes the different naming conventions between Vento-Tormo et al. and this manuscript. We hope this helps allay the concerns of the reviewers.

Also, evCTBs are included in the cells with significantly enriched recruited genes. Does this reflect contamination by these cells in analyses used to identify recruited genes in endometrial cells?

Given the low likelihood of contamination of placenta cells in our endometrial gene expression datasets (as shown in Table 1), it is likely that some genes that were recruited into endometrial stromal lineage cells were also recruited into placental cells. While some of these recruitment events may be by chance, it is also possible that pleiotropic gene regulatory elements evolved that mediate expression in both endometrial stromal and placental lineage cells. Previous studies have found that such correlated evolution of gene expression is pervasive in transcriptomes (PMCID: PMC5800078).

9) Figure 4A-C are meaningless to this reviewer. Large amount of data but little increment in knowledge.

Our use of a different nomenclature for endometrial stromal lineage cells than Vento-Tormo et al., and that recruited genes are preferentially enriched in endometrial stromal lineage cells led us to explore the development of this cell lineage in greater resolution. Figure 4 shows the developmental lineage of endometrial stromal lineage cells to justify our perivascular/ESF/DSC nomenclature, as well as highlight important biology of this cell lineage.

10) Please cite the source data for Figure 5A and B in the Figure legend. What is meant by minus 0 in the lower panel of Figure 5F? Figure 5F could be omitted.

The data for Figure 5A was generated by us in the study while the data shown in panel B is our re-analyses of the Vento-Tormo et al. dataset. Thus, we do not provide a citation for panel A, and have referenced that the source data for panel B is Vento-Tormo et al. in the main text.

We are unsure which figure the reviewer is referring to. While figure 5 does have a panel F, there is no minus 0 in the lower left.